METHODS

# ENQUIRE automatically reconstructs, expands, and drives enrichment analysis of gene and Mesh co-occurrence networks from context-specific biomedical literature

**Luca Musella[1]\*, Alejandro Afonso Castro[1], Xin Lai[1,2], Max Widmann[1,3], Julio Vera[1]\***

**1** Laboratory of Systems Tumor Immunology, Friedrich-Alexander-Universität Erlangen-Nürnberg, Deutsches Zentrum Immuntherapie, BZKF, and Uniklinikum Erlangen, Erlangen, Germany, **2** Faculty of Medicine and Health Technology, Systems and Network Medicine Lab, Biomedicine Unit, Tampere University, Tampere, Finland, **3** University of Konstanz, Konstanz, Germany

\* luca.musella@uk-erlangen.de (LM); julio.vera-gonzalez@uk-erlangen.de (JV)

**Data availability statement:** ENQUIRE's main program and the standalone scripts to perform the post hoc analyses are included in an Apptainer/Singularity image file (SIF), available for download at https://figshare.com/articles/software/ENQUIRE/24434845 (DOI: 10.6084/m9.figshare.24434845.v10), as well as in a Docker image available on Docker

## Abstract

The accelerating growth of scientific literature overwhelms our capacity to manually distil complex phenomena like molecular networks linked to diseases. Moreover, biases in biomedical research and database annotation limit our interpretation of facts and generation of hypotheses. ENQUIRE (Expanding Networks by Querying Unexpectedly Inter-Related Entities) offers a time- and resource-efficient alternative to manual literature curation and database mining. ENQUIRE reconstructs and expands co-occurrence networks of genes and biomedical ontologies from user-selected input corpora and network-inferred PubMed queries. Its modest resource usage and the integration of text mining, automatic querying, and network-based statistics mitigating literature biases makes ENQUIRE unique in its broad-scope applications. For example, ENQUIRE can generate co-occurrence gene networks that reflect high-confidence, functional networks. When tested on case studies spanning cancer, cell differentiation, and immunity, ENQUIRE identified interlinked genes and enriched pathways unique to each topic, thereby preserving their underlying context specificity. ENQUIRE supports biomedical researchers by easing literature annotation, boosting hypothesis formulation, and facilitating the identification of molecular targets for subsequent experimentation.

## Author summary

In biomedicine, we are interested in deciphering the molecular mechanisms underlying a disease, which are often governed by complex networks of interactions between macromolecules, like the proteins originated from the genes in our DNA. Discovering these networks requires aggregating data from several published scientific studies. However, this task entails considerable challenges, because each study usually just focuses on one or a few genes, and because the latter influence the processes in our body differently depending on the context they act in, such as a specific cell or clinical condition. On the

Hub at https://hub.docker.com/r/muszeb/enquire. Installation and running instructions can be found at https://github.com/Muszeb/ENQUIRE. Gene-symbol-to-alias lookup table, reference STRING network, reference *H. sapiens* Reactome pathways, input and output files from the case study, and data underlying the results are deposited at https://zenodo.org/records/12734778 (DOI: 10.5281/zenodo.12734778). All the individual scripts used in the ENQUIRE software are deposited as S1 Code.

**Funding:** This work was supported by the Bundesministerium für Bildung und Forschung (BMBF) through the projects e:Med – MelAutim I and II (grant numbers 01ZX1905A and 01ZX2205A to JV), which paid the salary of LM, KI-VesD I (grant number 161L0244A to JV), and KI-VesD II (grant number 16LW0338K to JV), which paid the salary of AAC. XL acknowledges funding from the Johannes and Frieda Marohn Foundation (grant Alz/Iko-Lai/2022, url: https://www.fau.de/glossary/johannes-und-frieda-marohn-stiftung) and the open access support funding from Tampere University, Finland. JV acknowledges funding from the Matthias Lackas Foundation (grant Berking/Vera-Gonzalez/Heppt/2021, url: https://lei.report/LEI/967600DXB06NITTF0831), which paid the salary of MW. The funders had no role in study design, data collection and analysis, decision to publish, or preparation of the manuscript.

**Competing interests:** The authors have declared that no competing interests exist.

one hand, manual literature search may result in an incomplete interaction network due to the difficulty in annotating all the relevant findings in an increasingly growing corpus of publications; on the other hand, established databases of molecular interactions omit the context in which the interaction has been observed.

Here, we present ENQUIRE (Expanding Networks by Querying Unexpectedly Inter-Related Entities), a software that offers a time- and resource-efficient alternative to manual literature search and database mining. ENQUIRE automatically parses biomedical articles from PubMed, a database of biomedical studies, starting from an initial set of relevant publications chosen by the user. It then extracts literature co-occurrences between genes and associated biomedical concepts – such as pathology and cell type – thus creating a network of co-occurring genes and concepts. It can also use genes and concepts that are central in the reconstructed network of co-occurrences to search for new PubMed articles and expand the original input corpus and, in turn, the network. Moreover, ENQUIRE-generated co-occurrence networks can complement previously annotated interactions by highlighting those that are relevant to the case study of interest, including previously undescribed ones.

ENQUIRE supports biomedical researchers by making literature annotation easy, boosting hypothesis formulation, and facilitating the identification of targets for laboratory experiments.

## Introduction

Curated gene networks are of high interest to prime the analysis of biomedical omics data, identification of disease-specific regulatory modules, and therapy-oriented studies like drug repurposing [1–4]. However, the growing biomedical literature corpus makes curation of biomolecular pathways challenging. Annotating molecular interactions from literature requires domain expertise, yet that same background knowledge could entail predispositions towards partial pictures of faceted biomedical problems [5]. In contrast, relation extraction from databases often omits the contextual information of gene interactions and can bias the results towards ubiquitously expressed, commonly investigated, and richly annotated genes [6–8]. This can make systematic comparisons of biomedical research topics inconclusive or unattractive from an expenditure perspective.

Recently, there have been significant investments in the automatic annotation of scientific corpora. The knowledgebase immuneXpresso indexes text-mined interactions among immune cells and cytokines [9], while SimText provides a framework to interactively explore the content of a user-provided corpus of literature [10]. These and other tools rely on natural language processing methods like named-entity recognition [11] (NER), part-of-speech recognition [12], directionality assignment [13], relationship detection, and co-occurrence scoring [14,15]. These efforts in biomedical text mining aim at detecting meta-features and co-occurrences in literature corpora. In particular, tools that reconstruct gene co-occurrence networks from literature can be divided into three main categories based on the framework used to infer relations. Methods such as Biblio-MetReS and MCforGN combine a NER system with bibliometric statistics, such that free-text documents can be parsed and a null model for observing one or multiple co-occurrences between a pair of genes by random chance can be designed [16,17]; while a probabilistic model allows to select co-occurrences via their statistical significance, the two predominantly used test statistics, contingency tables and

count statistics, inherently do not correct for either corpus size or the uneven occurrence distribution of individual entities. A second, non-strictly-probabilistic framework such as the one proposed in IMA and based on association rules can effectively assign weights based on different properties to an observed co-occurrence [18]; however, this comes at the cost of optimal parameter selection, which can differ between inputs, and would only be restricted to pre-annotated gene mentions if a database is used in place of a NER algorithm, as it's the case for IMA. A third strategy consists in deploying a part-of-speech recognition algorithm, dependency parser, or large language model trained on annotated gene-gene relations, at the individual-sentence level; For example, STRING deploys the RoBERTa language model and computes a confidence score for a text-mined co-occurrence using a weighting system that depends on the proximity between protein mentions in the free-text document [19,20]. However, this kind of framework does not assess the statistical significance and confidence level of a text-mined relation in dense, literature-based co-occurrence networks, as it does not incorporate the notion of co-occurrence by random chance nor the possibility of false or incorrect statements in text, irrespective of natural language processing [21,22]. We find this striking, as it's been established that observational studies need to be aggregated to produce stronger scientific evidence, especially in medicine, and that a finding in any such study may turn out to be irreproducible [23–26].

Alternative to co-occurrence networks are semantic networks, also known as knowledge graphs (KGs). In this type of data structure different attributes for nodes and edges are defined, functionally allowing the diversification of node interrelations. For instance, KGs have been successfully employed as graph databases to mine specific relationships of interest, like *Compound-binds-Gene* for drug repurposing, train machine learning models for the selection of novel candidate genes in antibiotic resistance, or classify text-mined and database knowledge in fixed semantic relations for assembling molecular models [27–29]. However, updating a KG or a model trained on it can require considerable effort and resources, to which the accelerating growth of scientific literature poses a challenge [30].

In an effort to overcome the above limitations, this study introduces ENQUIRE (Expanding Networks by Querying Unexpectedly Inter-Related Entities), an easily deployable software tool that performs automatic reconstruction and expansion of biomedical co-occurrence networks from a user-defined PubMed literature corpus. A distinctive element in our approach is a probabilistic framework that accounts for uneven representation of entities in the literature corpus without the biases of purely bibliometric test statistics. ENQUIRE applies a state-of-the-art random graph model to retrieve context-specific, significant co-occurrences, as a function of the occurrence count distribution of biomedical entities and the resulting total number of observed co-occurrences, independently of the literature corpus size [31,32]. ENQUIRE processes scientific articles by extracting Medical Subject Headings (MeSH) and gene mentions from article abstracts, thus enriching gene-gene co-occurrence networks with gene-MeSH and MeSH-MeSH relations. Unlike KGs, ENQUIRE-derived gene/MeSH interrelations do not possess semantic attributes, however they can be used in conjunction with available databases as inputs to *post hoc* analyses. As we show in a case study, these input-literature-specific co-occurrence networks can be used to construct gene sets described by the co-occurring MeSH terms and to score a reference physical interaction network according to the text-mined relations. These outputs provided by ENQUIRE can effectively be used to mine previously unannotated genes relevant to a topic of interest. In fact, several existing tools reconstruct gene networks from large, general purpose databases, thus requiring a set of so-called seed genes to filter and tailor the obtained networks to a topic of interest, such as a disease [33,34]; if a gene set is not available, further literature research would be needed. As an alternative to manual literature search, ENQUIRE can directly reconstruct a co-occurrence

network from a topic-restricted corpus, and also automatically generate PubMed queries from connected biomedical entities in the network, contextually expanding the original corpus and, in turn, the co-occurrence network.

To our knowledge, ENQUIRE is the first tool that integrates text mining, network reconstruction, automatic literature querying, and *post hoc* analysis. One of our goals was to offer a versatile, resource-efficient software that scaled well with arbitrarily complex case studies. In this study, we show ENQUIRE's efficiency, accuracy, and broad-scope applications in identifying relevant biomedical relations in different contexts and scenarios.

## Results

### A tool to generate co-occurrence networks from literature

ENQUIRE (Expanding Networks by Querying Unexpectedly Inter-Related Entities) is an algorithm that reconstructs and expands co-occurrence networks of *Homo sapiens* genes and biomedical ontologies (MeSH), using a corpus of PubMed articles as input. The method iteratively annotates MeSH and gene mentions from abstracts, it statistically assesses their importance, and retrieves further articles by generating network-informed PubMed queries. The algorithm iterates until a connected gene and MeSH network is obtained, unless a different exit condition that prevents ENQUIRE from continuing is encountered, as further described below. ENQUIRE can be described as a loop consisting of the following sequential steps (Fig 1):

a)  The user supplies an input literature corpus in the form of PubMed identifiers (PMIDs). An exit condition is triggered if less than three PMIDs are submitted;

b)  The algorithm fetches PubMed-indexed MeSH terms associated to the submitted PMIDs. Next, it retrieves and concatenates their titles, abstracts, and non-MEDLINE-indexed keywords ("Other Term") with respect to their PMIDs. Gene normalization, i.e. the standardization of all gene mentions to reference names, is performed using in-house designed tokenization, filtering by abstract-specific lists of ambiguous terms, and employing a lookup table of gene aliases. An exit condition is triggered if no entity can be retrieved;

c)  It then annotates individual co-occurrences between gene and MeSH entities and constructs a gene/MeSH undirected, heterogeneous multigraph;

d)  The method selects significantly over-represented co-occurrences and generates an undirected, simple graph. The test statistic is based on a random graph null model that provides an unbiased probability for the edge count between each node pair as a function of their sheer individual occurrence and total number of overall co-occurrences in the literature corpus. At the same time, the retained edges are weighted based on the specific node-node co-occurrence count across the literature corpus, using a probability function of the expected count for any node pair. An exit condition is triggered if no co-occurrence is statistically significant, resulting in an empty graph;

e)  Next, nodes are also weighted based on network centrality measures. In this newly weighted graph, the algorithm then selects "information-dense" maximal cliques, i.e. clusters of high-weight nodes all connected to each other, to perform network community detection on the corresponding nodes. An exit condition is triggered if no such clique can be found or if only one community exists;

f)  The algorithm identifies optimal sets of community-connecting graphlets via an approximate solution to the "travelling salesman problem" (TSP). An exit condition is triggered if no such community-connecting path can be constructed;

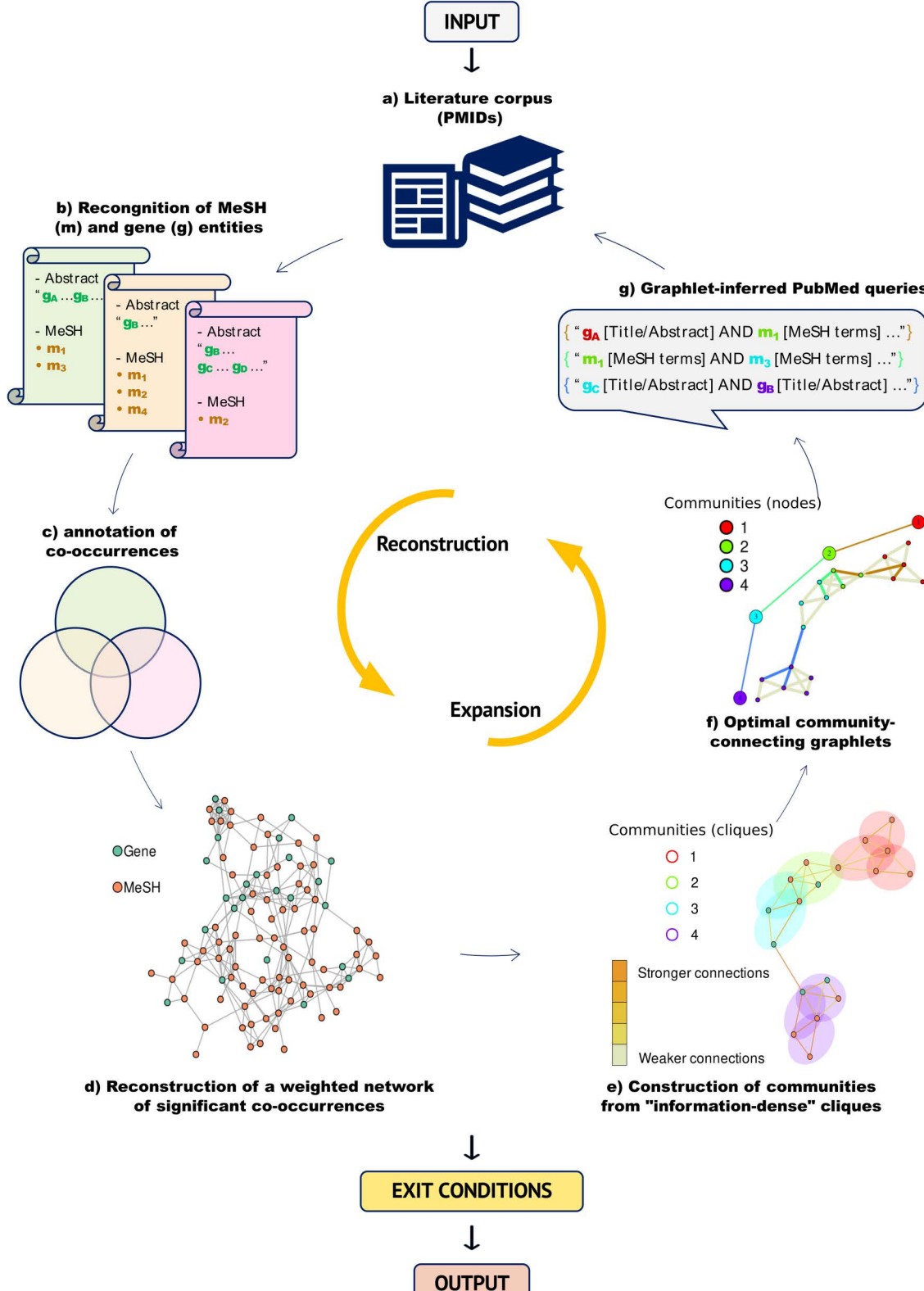

**Fig 1. Overview of ENQUIRE methodology.** ENQUIRE requires a set of PubMed identifiers as input. The pipeline iteratively orchestrates reconstruction and expansion of literature-derived co-occurrence networks, until an exit condition is met (see main text for details on exit conditions). Additional information about each alphabetically indexed module and output is provided in the Materials and methods section. For a more detailed flowchart see S1 Fig. We acknowledge the use of royalty-free Microsoft icons.

g) Finally, the algorithm translates the entity nodes corresponding to the identified community-connecting graphlets into PubMed queries to find additional, relevant articles. Should ENQUIRE find new articles, their PMIDs are joined with the previous ones and automatically provided to module b), starting a new iteration. Otherwise, the algorithm terminates.

Whenever ENQUIRE reconstructs a network from the union of old and new PMIDs, the previously reconstructed network is joined with the new one. The joined network has recomputed edge and node weights in accordance to its expanded literature corpus and connectivity. The rationale is to prioritize the original reconstruction, while also leveraging the expanded literature corpus.

Users can tune five options to tailor the workflow, namely:

1) Entity scope $e$ to restrict target entities to genes or MeSH only, in case only one entity type is of interest for the research scope – default: both;

2) representativeness threshold $t$ to disregard subgraphs reconstructed at step (d) and characterized by poor overlap with the literature corpus, in order to run the expansion step using only the most evidence-supported graphs – default: 1% overlap, that is nodes in such subgraphs are mentioned in at least 1% of the articles;

3) attempts $A$ to choose the number of consecutive TSP solutions aimed at connecting network communities at step (f), thus increasing or decreasing the chances of finding new PMIDs – default: 2 attempts, the second of which is computed after removing the previously found community-connecting graphlets from the network;

4) query size $k$ to control the number of entities that compose community-connecting graphlets at step (f) and that must be simultaneously used in a PubMed query at step (g), thus controlling the query strictness – default: 4 entities are queried at once by "AND" concatenation;

5) connectivity criterion $K$ to exclude newly found entities not having edges with nodes from $K$ communities previously generated at step (e), thus limiting the number of newly found nodes and edges – default: 3 communities.

ENQUIRE's goal is to generate a gene/MeSH network and its respective gene- and MeSH-only subgraphs that individually consist of a single, connected component. With default parameters, ENQUIRE outputs node and edge lists of a gene/MeSH co-occurrence network and the respective gene- and MeSH-only subgraphs at each iteration. The final ENQUIRE results include graphics, summary statistics, and tabulated co-occurrences paired to literature supporting any putative relation, as well as to known Reactome-annotated interactions in the case of gene-gene co-occurrences, allowing subsequent verification and *post hoc* analyses. For instance, it is possible to extract subsets of the literature corpus that support a gene/MeSH relation of interest and access the articles via hyperlinks redirecting to PubMed via the TSV file with suffix *Complete_edges_literature_links.tsv*. See Materials and methods, S1 Fig, and https://github.com/Muszeb/ENQUIRE for a comprehensive description of the algorithm and its parameters.

## A case study using ENQUIRE

To showcase ENQUIRE, we set up a small-scale case study in which we looked for literature-based relationships between the immune system and ferroptosis, a form of programmed cell death [35]. We selected 27 papers obtained from the PubMed query *("Ferroptosis"[MeSH terms] AND "Immune System"[MeSH terms]) NOT "review"[Publication Type]"* – queried

on 14.04.23. We refer to this case study as *Ferroptosis and Immune System*. We increased the number of attempts $A$ to 3 and decreased the connectivity criterion $K$ to 2, as we expected few query-matching PMIDs. The expansion process is depicted in Fig 2, using the Cytoscape package DyNet [36,37]. The original reconstructed network consists of four connected components. The first expansion led to additional, significant co-occurrences and newly found entities that connected the four components into a single one. The algorithm stopped after obtaining a single, connected gene/MeSH network and not finding additional query-matching PMIDs. Using up to 6 CPU cores, ENQUIRE finished in 16 minutes using less than 0.4 GB of RAM (S2 Fig). Two graphlets resulted in PMID-matching queries in the first expansion, and three more generated non-empty queries in the second expansion, overall increasing the original corpus by six PMIDs. The topology of newly found nodes and edges demonstrates that the network expansion process does not exclusively involve the direct neighborhood of the query-generating graphlets, but also distant and previously disjoint network communities (Fig 2).

Next, we applied context-specific gene set annotation on the *Ferroptosis and Immune System* original gene/MeSH co-occurrence network, as described in Materials and methods. We identified gene sets that capture meaningful and biologically relevant associations (Fig 3A), whose Fuzzy-c-means-derived centroids suggest a contextual separation of the studied genes based on involved immune cells (T lymphocytes, macrophages, and neutrophils), pathology (melanoma, glioblastoma, colonic and breast neoplasms), and metabolism (lipid metabolism and autophagy). To further showcase the additional benefits of ENQUIRE-reconstructed, literature-supported gene sets, we compared their overlap to existing annotated gene sets. First, we applied over-representation analysis (ORA) to the set of protein-coding genes extracted in the original network reconstruction (Fig 3A), setting the entire *H. sapiens* proteome as *universe*. STRING's in-built ORA tool returned, among many, *Regulation of glutathione biosynthetic process*, *Intracellular metabolism of fatty acids regulates insulin secretion*, and *Ferroptosis* as the most significantly-enriched gene sets and pathways, in order of log-odds, respectively annotated under the Gene Ontology, Reactome, and KEGG databases (https://zenodo.org/records/12734778). In contrast, no Reactome pathway was found to be significantly over-represented in any reconstructed gene set, using the complete set of genes from the original network reconstruction as *universe* (https://zenodo.org/records/12734778).

We also applied context-aware pathway enrichment analysis to the *Ferroptosis and Immune System* gene-gene co-occurrence subgraphs with the approach described in Materials and methods. We summarized the results in Fig 3B, which depicts 30 Reactome pathways whose adjusted p-values were below 5% FDR for at least one network, sorted by Reactome category. In the original network, we obtained enrichments of pathways centered around Toll-like receptor and MAP kinases signaling cascades (e.g. R-HSA-975138). In the expanded networks, the metabolic pathway *Glutathione conjugation* (R-HSA-156590) and additional innate immunity-related and programmed cell death pathways were enriched.

ENQUIRE performs this topology-based pathway enrichment by assigning node weights derived from a gene-gene co-occurrence network (Q-scores) to protein-coding genes of a reference physical interaction network (STRING). We can therefore further investigate text mining-inferred network communities in the weighted reference network to potentially discover novel or more comprehensive pathways. We then applied the InfoMap community detection algorithm to Q-score-weighted and unweighted variants of the same reference STRING network [38]. Upon normalization using the empirical cumulative density function, the Q-scores inferred from the *Ferroptosis and Immune System* original gene-gene co-occurrence network introduced a bias toward high-weighted nodes in InfoMap's random walk model. 23 out of 724 communities differed by at least one node from any

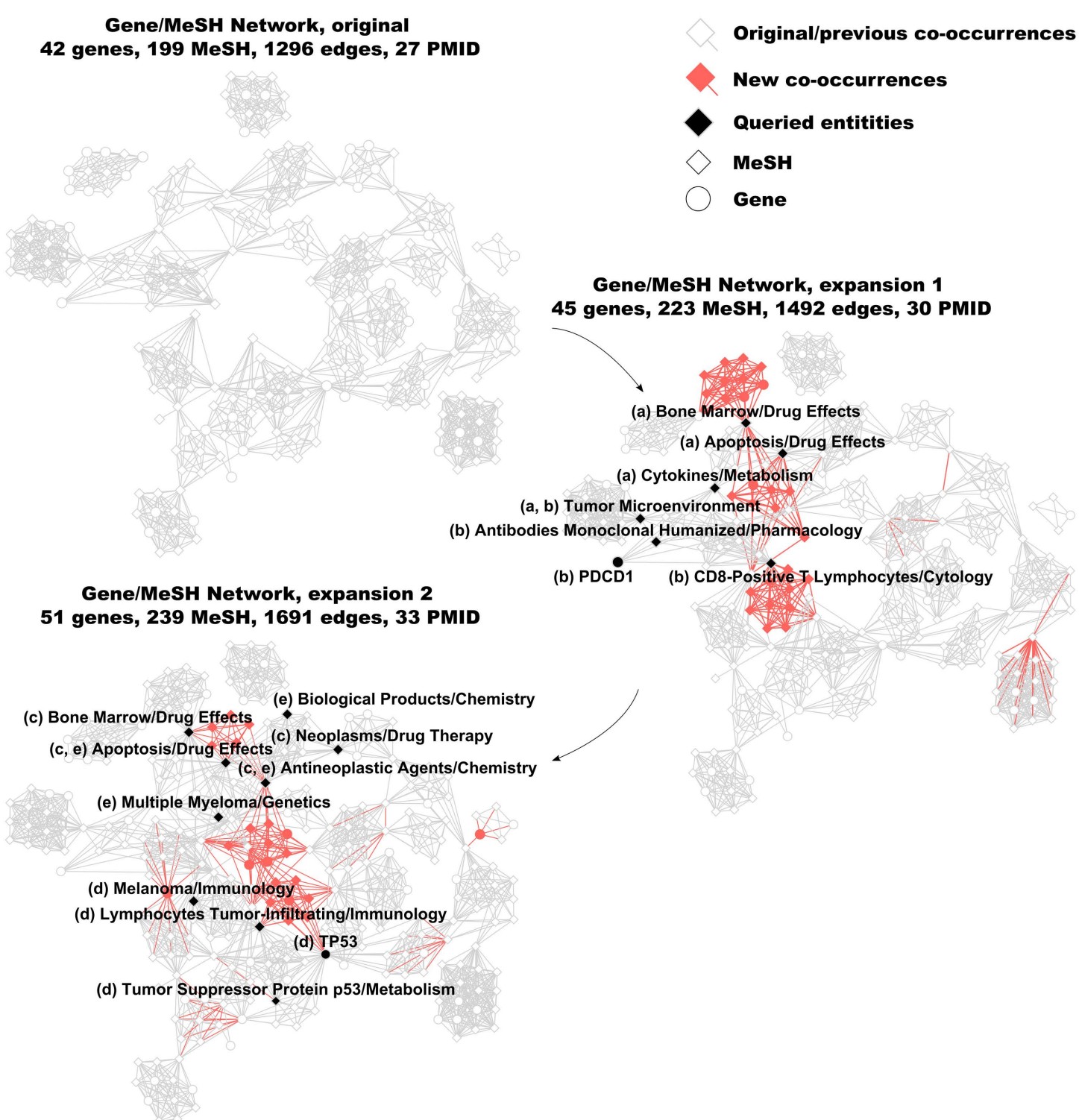

**Fig 2. Example of ENQUIRE's network reconstruction and expansion.** We generated co-occurrence networks using the corpus collected for the case study *Ferroptosis and Immune System* as input (see main text for additionally specified parameters). The originally reconstructed network and the expanded ones obtained by querying community-connecting graphlets are arranged clockwise. Nodes and edges belonging to previously reconstructed networks are colored in white and grey, respectively. At each network expansion, newly found nodes and edges are indicated in red. Nodes of the five graphlets that resulted in PMID-matching queries are colored in black and labelled, with letters in parentheses (a-e) indicating the graphlets they belong to. We acknowledge the use of Cytoscape and DyNet to layout the networks.

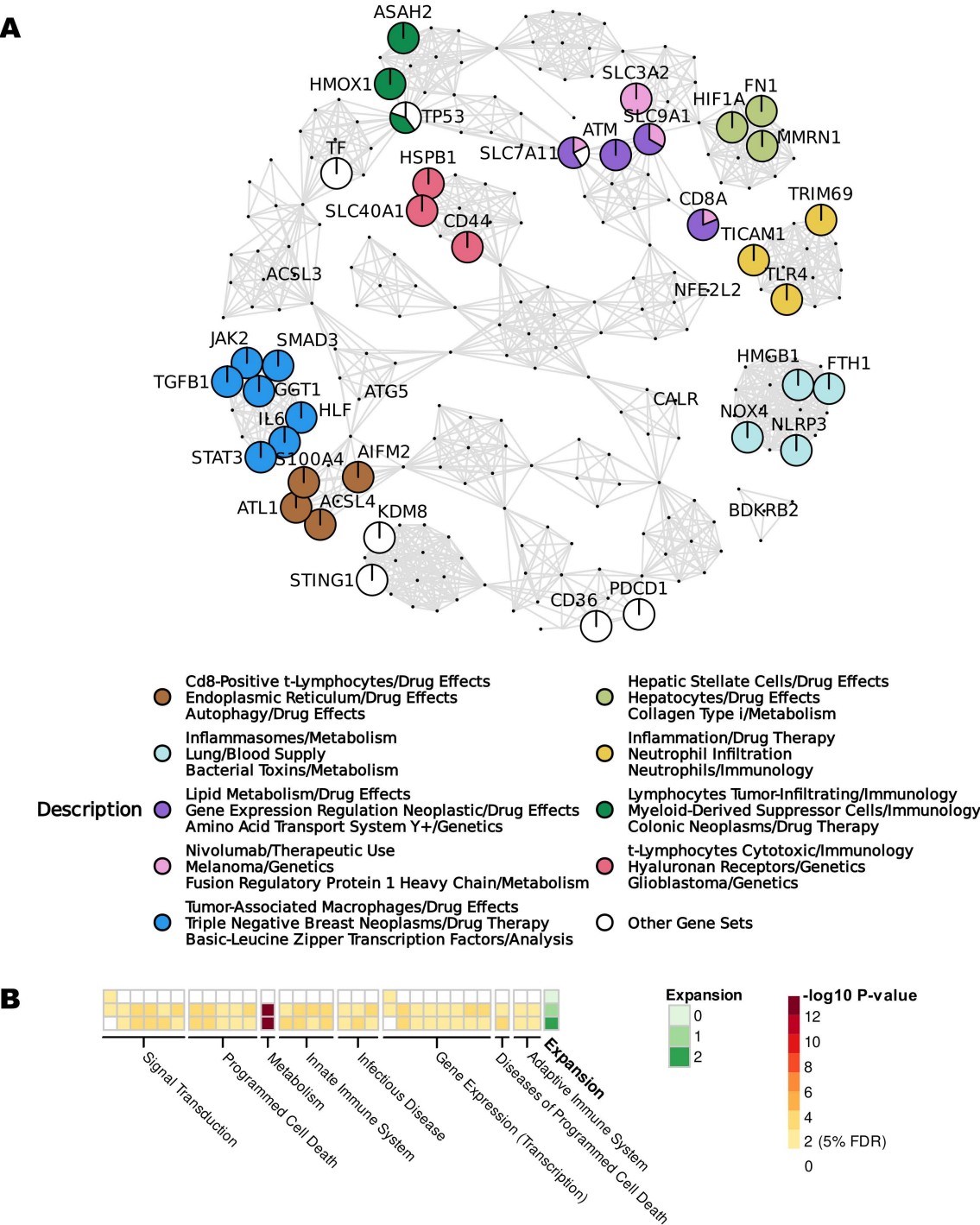

**Fig 3. Example of ENQUIRE's *post hoc* analyses.** We used the PubMed identifiers (PMIDs) generated by the query *("Ferroptosis"[MeSH terms] AND "Immune System"[MeSH terms]) NOT "review"[Publication Type]* as input and obtained ENQUIRE-reconstructed gene and MeSH co-occurrence networks. **A**: output of the automatic gene set reconstruction, using the original gene/ MeSH network as input and fuzzy c-means. Nodes referring to genes are labelled, and those belonging to clusters containing 2 or more genes are represented as pie charts. Sector sizes of the pie-chart-shaped nodes reflect their relative membership degree to each gene set cluster. For simplicity, a color legend and description are provided only for gene sets of size 3 or bigger. **B**: topology-based enrichment analysis of Reactome pathways, using original and expanded networks, as described in Materials and methods. 30 pathways whose FDR-adjusted p-value was significant in at least two networks are depicted. Reactome pathways are grouped based on "Top-Level Pathway" and "Disease" categories. Green and white-yellow-red gradients respectively indicate the expansion counter and observed, unadjusted p-values.

community identified in the unweighted variant of the STRING network (https://zenodo.org/records/12734778). We illustrated the three weighted-variant-exclusive communities with largest average Q-score in S3 Fig; these communities show physical interrelation via the genes SDC2, TGFB1, BGN, CD36, and LYN, among others. ORA conducted on the Reactome database revealed that most communities overlap with several previously annotated pathways, such as MAP kinase, Toll-like receptor, and TGF-beta signaling, similarly to what described above. Only one InfoMap-derived community did not significantly overlap with any previously annotated pathway, which includes the positive-scored gene IL6ST, as well as the unscored genes LIF, and LIFR. This lack of overlap prompted us to conduct a literature search in which we discovered a LIFR-centered molecular axis vulnerable to ferroptosis in ovarian and liver neoplasms among sources not text-mined by ENQUIRE [39,40].

In summary, our case study on *Ferroptosis and Immune System* highlights potential molecular axes between iron-regulated cell death, pathophysiology, metabolism, and immune response [41–43]. The reconstructed gene sets suggest a separation mainly based on disease studied and immune cell entities. Additional analysis on communities in the Q-score-weighted physical interaction network reveals a further molecular-axis tangential to those found by pathway enrichment analysis. This ferroptosis-regulating axis has been recently investigated as a potential therapeutic target in cancer [39,40]. Together with a structured output reporting literature references that support significant co-occurrences, these results showcase ENQUIRE's distinctive integration of text mining, network-based statistics, and *post hoc* analysis on pre-annotated databases. This approach can lead to enrichment analyses for omics data with previously unannotated gene sets, generation of molecular models based on literature evidence, and annotation of more comprehensive or previously unannotated pathways.

## ENQUIRE's gene normalization favors precision and resource efficiency

ENQUIRE is intended to consume abstracts from studies in *H. sapiens* and *Mus musculus*. We therefore evaluated ENQUIRE's precision and recall using the abstracts in the NLM-Gene corpus mentioning at least one *M. musculus* or *H. sapiens* gene – 479 out of 550 entries [44]. The gene normalization task is here defined as detecting at least one gene alias per unique reference gene mentioned in an abstract. We ran the computations on a Linux computer with 20 CPUs (3.1 GHz) and 252 GB of RAM. Up to 8 cores were used for parallelization. ENQUIRE's maximum F1 score is 0.747, corresponding to 0.822 precision and 0.683 recall, using as little as 0.36 GB of RAM and with speeds up to 0.03 seconds per abstract (Table 1). The Schwartz-Hearst abbreviation-definition detection algorithm improves precision of tokenization and normalization by 2%, without major loss in recall nor higher computational requirements [45]. In some use cases, it could be necessary to exclude gene mentions associated to cell entities, such as "CD8+ lymphocytes". The scispaCy's *en_ner_jnlpba_md* model removes unwanted gene-matching cell mentions, at the cost of about 2% reduction in recall [46]. It should be noted, however, that the latter metric is affected by the fact that gene mentions included in cell entities are counted as true positives in the NLM-Gene corpus.

We also compared ENQUIRE's performance to GNorm2, a state-of-the-art deep-learning model for gene entity recognition and normalization [47]. We tested ENQUIRE's most resource-intensive configuration (both *en_ner_jnlpba_md* and Schwartz-Hearst modules enabled) against GNorm2's implementation of Bioformer, a state-of-the-art deep-learning model based on BERT, but 60% smaller in size, and whose observed precision and recall for the normalization task were 0.834 and 0.726, respectively [48]. Table 2 shows that GNorm2 is considerably slower and has a higher resource usage than ENQUIRE. If ENQUIRE were to implement GNorm2 for gene normalization, this would impair its usage in scenarios with

**Table 1. Performance of ENQUIRE's gene normalization algorithm.** Precision, recall, and their harmonic mean (F1) are based on 479 abstracts from the NLM-Gene corpus containing at least one mention to a *H. sapiens* or *M. musculus* gene. Different gene normalization methods were evaluated by adding or removing filters for excluding predicted cell entities (*en_ner_jnlpba_md*) and ambiguous abbreviation-definition pairs (Schwartz-Hearst). Gene mentions contained in cell entities such as "CD8+ T cell" are true positives in the NLM-Gene corpus. Text spans tagged as cell entities by the *en_ner_jnlpba* model are removed without being processed by the tokenizer module. Maximum RAM usage is measured as resident set size (RSS). Estimated time in seconds per abstract (sec/abstract) also accounts for loading the gene alias lookup table and machine learning models. The best values for each parameter setting are highlighted in bold.

| Gene normalization Method | Precision | Recall | F1 | Computing performance | | | |
|---|---|---|---|---|---|---|---|
| | | | | Resource usage | Threads | | |
| | | | | | 1 | 4 | 8 |
| *en_ner_jnlpba_md* + Schwartz-Hearst + ENQUIRE tokenizer/dictionary | **0.823** | 0.662 | 0.734 | Max. RSS (GB) | 1.95 | 1.95 | 1.95 |
| | | | | sec/abstract | 0.172 | 0.0656 | 0.0488 |
| **Schwartz-Hearst + ENQUIRE tokenizer/dictionary** | 0.822 | 0.683 | **0.747** | Max. RSS (GB) | **0.359** | **0.359** | 0.361 |
| | | | | sec/abstract | 0.125 | 0.0435 | 0.0318 |
| *en_ner_jnlpba_md* + ENQUIRE tokenizer/dictionary | 0.804 | 0.666 | 0.728 | Max. RSS (GB) | 1.95 | 1.95 | 1.95 |
| | | | | sec/abstract | 0.148 | 0.0651 | 0.0481 |
| **ENQUIRE tokenizer/dictionary** | 0.802 | **0.688** | 0.741 | Max. RSS (GB) | 0.360 | **0.359** | **0.359** |
| | | | | sec/abstract | **0.105** | **0.0400** | **0.0280** |

**Table 2. Differences in computing performance between ENQUIRE's gene normalization algorithm and GNorm2-Bioformer.** We ran the computations on a Linux computer with 20 CPUs (3.1 GHz) and 252 GB of RAM. Up to 8 cores were used for parallelization. Maximum RAM usage was measured as resident set size (RSS). Estimated time in seconds per processed abstract (sec/abstract) also accounts for loading gene alias lookup tables and machine learning models.

| Gene normalization method | Corpus size | Computing performance | | | |
|---|---|---|---|---|---|
| | | Resource usage | Threads | | |
| | | | 1 | 4 | 8 |
| *en_ner_jnlpba_md* + Schwartz-Hearst + ENQUIRE tokenizer/dictionary | 26 | Max. RSS (GB) | 1.95 | 1.95 | 1.95 |
| | | sec/abstract | 0.573 | 0.509 | 0.513 |
| **GNorm2-Bioformer** | | Max. RSS (GB) | 17.3 | 16.4 | 17.4 |
| | | sec/abstract | 4.310 | 4.150 | 2.73 |
| *en_ner_jnlpba_md* + Schwartz-Hearst + ENQUIRE tokenizer/dictionary | 130 | Max. RSS (GB) | 2.08 | 1.95 | 1.95 |
| | | sec/abstract | 0.205 | 0.134 | 0.125 |
| **GNorm2-Bioformer** | | Max. RSS (GB) | 25.1 | 25.1 | 24.7 |
| | | sec/abstract | 2.500 | 1.260 | 1.070 |
| *en_ner_jnlpba_md* + Schwartz-Hearst + ENQUIRE tokenizer/dictionary | 1300 | Max. RSS (GB) | 5.9 | 2.91 | 2.71 |
| | | sec/abstract | 0.118 | 0.044 | 0.030 |
| **GNorm2-Bioformer** | | Max. RSS (GB) | 25.0 | 24.8 | 24.9 |
| | | sec/abstract | 2.370 | 1.050 | 0.835 |

limited resources and computing time: for example, we verified that GNorm2 cannot be run on the CPU-based computer with 16 GB of RAM used for the case study in Fig 2 (see https://zenodo.org/records/12734778). In these terms, ENQUIRE's *in-house* gene normalization is more suitable for text mining large input corpora on a variety of devices beyond CPU-based computer clusters.

## ENQUIRE networks support ranking of genes relevant to the input literature

To evaluate ENQUIRE's ability in inferring genes relevant to the input corpus, we extracted *H. sapiens* pathways, their belonging genes, and corresponding primary literature references from the Reactome Graph Database [49]. We used the lists of references as inputs

and performed a single gene entity-restricted co-occurrence network reconstruction for each pathway. Out of 967 examined pathways, ENQUIRE successfully reconstructed a gene co-occurrence network from the reference literature of 720 of them. We evaluated the effect of input corpus size, pathway size and average entity co-occurrence per paper on the accuracy of the resulting networks (Table 3). As expected, precision and recall show opposite Spearman's correlation trends concerning corpus and pathway sizes, but average gene-gene co-occurrence per article appears uncorrelated. The negative correlation between corpus size and precision is −0.18, suggesting a low impact of large input corpora on the output.

Next, we investigated if the ENQUIRE-computed weight $W$, an aggregated measure of network centrality and literature support of its connections, is a useful measure of gene relevance regarding the input corpus (Materials and methods). To this end, we analyzed the above-mentioned gene-scope co-occurrence networks. In Fig 4, we compared the pan-pathway-aggregated distributions of ENQUIRE-derived, true positive (top panel) and false positive (middle panel) genes as a function of $W$ (x-axis). We subdivided the distribution into four evenly spaced intervals, performed a chi-square test of independence, which resulted to be significant, and extracted the standardized Pearson residuals for true positives and false positives (colored boxes beneath the distributions). True positives tend to have higher node weights than false positives. Nodes having weights higher than 0.75 are overrepresented in the true-positive distribution, as indicated by the Pearson residuals. This suggests one can use the node weight $W$ to rank a set of ENQUIRE-derived genes based on their relevance to the literature corpus in question.

## ENQUIRE recovers genes with high chances of reflecting biochemical interrelations

We hypothesized that ENQUIRE-derived gene co-occurrence networks could be enriched in molecular gene-gene interactions annotated in databases. To test this, we queried PubMed with all possible cross-pairs of *Diseases* and *Genetic Phenomena* MeSH terms. We further processed the 3098 queries that retrieved 50–500 matching PMIDs and extracted their gene-gene co-occurrence networks obtained after one network reconstruction. We then inspected whether their respective protein-coding genes can produce significant functional association networks based on STRING's protein-protein interaction (PPI) database (see Materials and methods) [50]. Table 4 indicates that for 1336 (43.1%) MeSH pairs, both ENQUIRE and STRING generated a minimal network with at least three genes and two edges. In a subset of 733 networks with degree sequences allowing at least ten different graph realizations, we assessed ENQUIRE's capability of reflecting functional interactions. We then generated two empirical random probability distributions for STRING's edge count and DeltaCon similarity score [51] (see Materials and methods). Within the tested networks, 730 protein-coding gene networks (99.6%) produced a STRING network with a higher edge count than 95% of equal-sized random STRING networks (PPI score). At the same time, 439 networks (59.9%) showed concordance with STRING-derived PPI networks based on statistically significant DeltaCon

**Table 3. Effect of relevant covariates on quality indicators of ENQUIRE's gene entity recognition.** We evaluated the effect of corpus size (input), Reactome pathway size (number of genes to be retrieved), and average gene-gene co-occurrence per PMID, using Spearman's correlation coefficients, for each measure. Bold indicates significant correlations, based on adjusted, Edgeworth-series-approximated p-values (See also https://zenodo.org/records/12734778).

| Metric | Corpus Size | Pathway Size | Average co-occurrence |
|---|---|---|---|
| **Precision** | **−0.18** | **0.49** | −0.06 |
| **Recall** | **0.46** | **−0.35** | **0.14** |

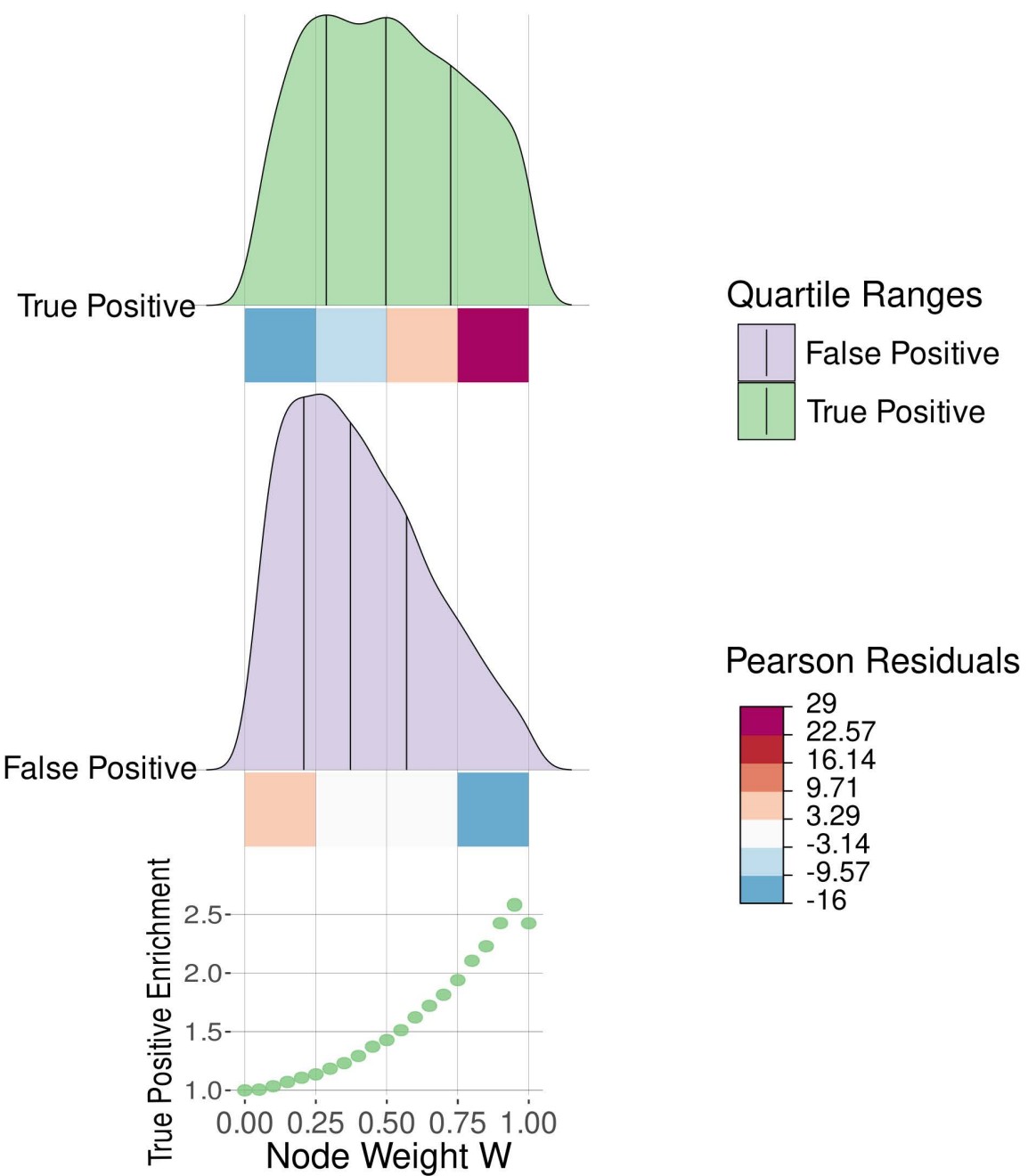

**Fig 4. Node weight distribution of ENQUIRE-derived gene networks correlate with relevance to the input literature corpus.** We defined true and false positives genes according to their presence or absence in a Reactome pathway, whose reference literature was used to retrieve gene mentions via ENQUIRE's gene normalization and network reconstruction. The statistics shows the aggregated results from 720 Reactome-derived input corpora. The aggregated distributions for true and false positive genes are segmented into quartiles. We defined four ranges of the node score $W$, indicated by squares, whose colors reflect Pearson standardized residuals resulting from a significant chi-square statistic. The lower chart depicts the enrichment of true positive genes, after pruning ENQUIRE-derived networks based on different values of $W$. Values are relative to the original proportion of true positives.

**Table 4. Relevant quality indicators of functional associations in 3098 case studies.** Percentages reported for edge count and DeltaCon significance independently refer to the set of 733 ENQUIRE-derived, tested networks, i.e., those with 10 or more possible realizations of the same degree sequence.

| Property | Subset | | Raw count | Percentage over the preceding step | Percentage over total (3098) |
|---|---|---|---|---|---|
| Network topology | At least 3 genes and 2 edges in both ENQUIRE and STRING networks | | 1336 | / | 43.1% |
| | At least 10 possible realizations of the same degree sequence | | 733 | 54.9% | 23.7% |
| Significance | Edge count p-value | < 0.05 | 730 | 99.6% | 23.6% |
| | | < 1% FDR | 722 | 98.5% | 23.3% |
| | DeltaCon p-value | < 0.05 | 439 | 59.9% | 14.2% |
| | | < 1% FDR | 344 | 46.9% | 11.1% |

similarities. After p-value adjustment, 722 (98.5%) and 344 (46.9%) ENQUIRE networks still show significantly high PPI scores and DeltaCon similarities, respectively.

To evaluate the effect of network size, we subdivided the 733 suitable networks into quartiles based on their node number and mapped the respective unadjusted p-value distributions of the above-described test sets. The edge-count-associated significant p-values increased with network size (Fig 5A). At the same time, the observed DeltaCon similarity values monotonically decrease with network size (Table 5). This is in accordance with DeltaCon's implementation of edge importance and zero-property [51], as differences in edge counts and number of connected components between ENQUIRE and STRING increase with the number of nodes. Nevertheless, we did not find a negative correlation between network size and p-values of observed DeltaCon similarities; instead, the quartile corresponding to the largest networks also shows the largest relative proportion of significant, adjusted p-values (Fig 5B). Taken together, our results suggest that ENQUIRE generates networks that frequently contain established, high-confidence functional relations. At the same time, ENQUIRE can potentially uncover gene interactions that were not previously annotated in databases, which can be directly verified by accessing the PMIDs reported in the output edge tables.

## ENQUIRE improves the context resolution of topology-based pathway enrichment analyses

We also analyzed ENQUIRE's ability to generate and expand co-occurrence networks with distinctive biological and biomedical signatures by literature querying. In particular, we evaluated the context resolution of ENQUIRE-generated gene networks, i.e. their ability to preserve differences and similarities in gene mention content from different corpora. To this end, we applied the complete ENQUIRE pipeline with default parameters to a comprehensive set of case studies, spanning cancer, cell differentiation, innate immunity, autoimmune diseases, and a positive control (Table 6).

Notice that each case study's input corpus is a perfect subset of the positive control corpus, which corresponds to a Szymkiewicz-Simpson overlap coefficient (OC) of 100% - see Materials and methods. Despite that, the positive control network does not necessarily exhibit an OC of 100% with non-expanded networks, in terms of both nodes and edges (S4 Fig). This demonstrates that ENQUIRE's network reconstruction is sensitive to the input corpus. Fig 6A depicts the expected dendrogram of the different case studies and respective expansions, based on their major topics and original input corpora. Fig 6B shows the observed clustering based on using ENQUIRE-informed, topology-based pathway enrichment analysis using the $Q$ score and SANTA's KNet [52] (see Post Hoc Analyses in Materials and methods and

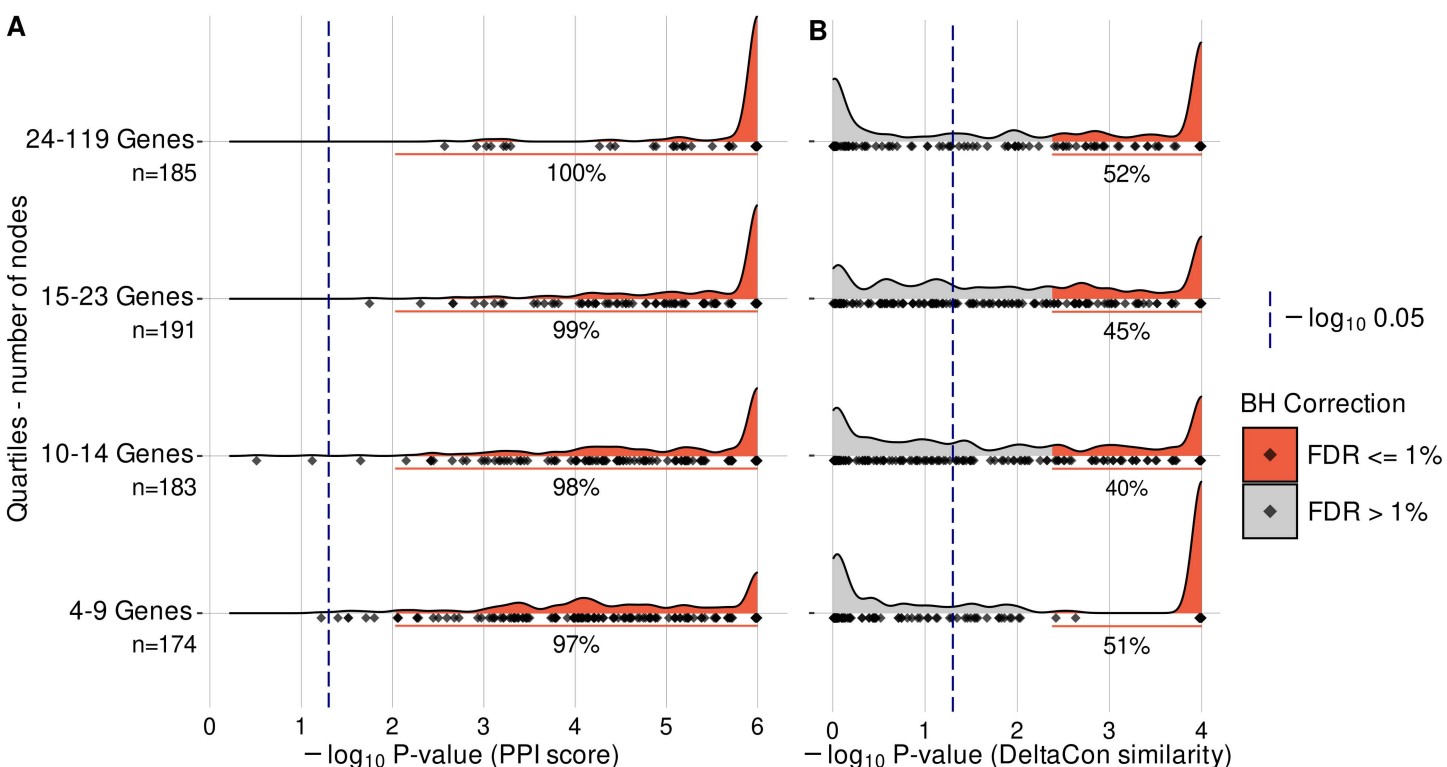

**Fig 5. Protein-coding genes from ENQUIRE-generated graphs significantly share functional associations.** Panels **A** and **B** respectively report the unadjusted p-value density distributions of STRING-informed edge counts and DeltaCon similarities, arranged by number of protein-coding genes (network size). We used the *H. sapiens* functional association network from STRING to evaluate ENQUIRE-derived networks of protein-coding genes. We tested 733 networks having 10 or more possible network realizations given the observed degree sequence. For each observed network size and degree sequence of ENQUIRE-generated gene networks, 1,000,000 and 10,000 samples were respectively generated to perform a test statistic on the observed edge counts and DeltaCon similarities. See Materials and methods for additional information. The 733 tested networks are apportioned into quartiles based on network size, and for each the exact size is indicated (n). Within each network size interval, grey and red areas respectively highlight insignificant and significant p-values with respect to a globally-applied Benjamini-Hochberg correction (BH), and a percentage is indicated for those below 1% FDR. Diamonds indicate the observed data.

S5 Fig). The 50 pathways with at least one significant, adjusted p-value (5% FDR) and highest p-value variances across case studies are depicted. The heat-map suggests that the case studies primarily cluster based on the affinities between their major topics, in agreement with the expected dendrogram. For example, pathways categorized under *Diseases of Metabolism*, *Diseases of Immune System*, and *Innate Immune System* are predominantly enriched in networks originated from the case study "Macrophage's signal transduction during M. tuberculosis infection" (MP-ST) and the major topic "Antigen Presentation in Autoimmune Diseases". Similarly, some of *Chromatin Organization* and *Developmental Biology* pathways are almost exclusively enriched in the networks corresponding to oligodendrocyte differentiation. Interestingly, a set of pathways linked to cell cycle like *Cyclin D associated events in G1* (R-HSA-69231) and enriched in the oligodendrocyte case study are reported to be also relevant in glioblastoma [53–56]. All case studies appear constitutively enriched in a cluster of *Pathways in Cancer* annotated downstream of *Diseases of signal transduction by growth factor receptors and second messengers* (R-HSA-5663202). We investigated this potential limitation in context-resolution and found that i) binned network distances between genes in R-HSA-5663202 subpathways employed by KNet are not significantly smaller than those within other tested pathways; ii) Spearman correlations between p-values and network or corpus sizes are

**Table 5. Empirical quantiles of DeltaCon similarities, ENQUIRE- and STRING-based edges counts, sorted by number of genes in the network.** Median values with respect to each metric and range of gene counts are highlighted in bold.

| Metric | Range of gene counts | Quantiles | | | | |
|---|---|---|---|---|---|---|
| | | 0% | 25% | 50% | 75% | 100% |
| **DeltaCon** | **4–9** | 0.75 | 0.83 | **0.87** | 0.94 | 1.00 |
| | **10–14** | 0.67 | 0.78 | **0.81** | 0.83 | 1.00 |
| | **15–23** | 0.65 | 0.74 | **0.77** | 0.79 | 0.87 |
| | **24–119** | 0.56 | 0.65 | **0.69** | 0.72 | 0.81 |
| **Edge count - ENQUIRE** | **4–9** | 4 | 6 | **7** | 8 | 16 |
| | **10–14** | 6 | 8 | **10** | 13 | 43 |
| | **15–23** | 8 | 13 | **17** | 22 | 66 |
| | **24–119** | 18 | 36 | **49** | 77 | 295 |
| **Edge count - STRING** | **4–9** | 4 | 6 | **8** | 10 | 23 |
| | **10–14** | 6 | 11 | **15** | 20 | 50 |
| | **15–23** | 10 | 21 | **28** | 37 | 94 |
| | **24–119** | 19 | 54 | **89** | 146 | 591 |
| **Connected components - ENQUIRE** | **4–9** | 1 | 2 | **2** | 3 | 5 |
| | **10–14** | 1 | 3 | **4** | 5 | 8 |
| | **15–23** | 1 | 4 | **6** | 7 | 12 |
| | **24–119** | 1 | 4 | **6** | 7 | 15 |
| **Connected components - STRING** | **4–9** | 1 | 1 | **2** | 2 | 5 |
| | **10–14** | 1 | 2 | **2** | 3 | 6 |
| | **15–23** | 1 | 2 | **3** | 4 | 8 |
| | **24–119** | 1 | 2 | **2** | 4 | 12 |

**Table 6. Selection of case studies for assessment of context resolution at the molecular pathway level.** We obtained PubMed queries by "AND" concatenation of up to three MeSH terms and further filtered to retrieve review articles only. The "Corpus size" refers to the non-redundant union of publications cited by three independent review articles, reported under the "References" column.

| Major Topic | Case Study (abbreviation) | PubMed Query | | | Corpus size | References |
|---|---|---|---|---|---|---|
| | | MeSH 1 | MeSH 2 | MeSH 3 | | (PMID) |
| Signal transduction in solid tumors | Melanoma (MM-ST) | Signal transduction | Melanoma | | 944 | 25587943, 32605090, 34924562 |
| | Uveal melanoma (UM-ST) | Signal transduction | Uveal neoplasms | | 218 | 25296731, 25113308, 28223438 |
| | Colorectal cancer (COL) | Signal transduction | Colorectal neoplasms | | 556 | 34884633, 34742312, 35836256 |
| | Breast cancer (BRE-ST) | Signal transduction | Breast neoplasms | | 522 | 29455658, 31752925, 32245065 |
| Macrophage's signal transduction in disease | Macrophage signal transduction upon infection (MP-ST) | Signal transduction | Macrophages | Mycobacterium tuberculosis | 470 | 32849525, 33558322, 34502407 |
| | Tumor-associated Macrophages (MP-TA) | Signal transduction | Tumor associated macrophages | | 386 | 33365025, 35844605, 35740975 |
| Antigen presentation in autoimmune diseases | Inflammatory bowel disease (IBD-AP) | Antigen presentation | Inflammatory bowel diseases | | 445 | 28534191, 33584726, 33800865 |
| | Rheumatoid arthritis (RA_AP) | Antigen presentation | Arthritis, rheumatoid | | 452 | 27225300, 28451787, 30589082 |
| | Psoriasis (PSO-AP) | Antigen presentation | Psoriasis | | 435 | 26215033, 29316717, 33050592 |
| Oligodendrocyte differentiation | Oligodendrocyte (ODC) | Cell differentiation | Oligodendroglia | | 355 | 24979526, 30770136, 31614602 |
| Positive control | All case studies (CTR) | All queries ("OR" concatenation) | | | 3606 | All of the above |

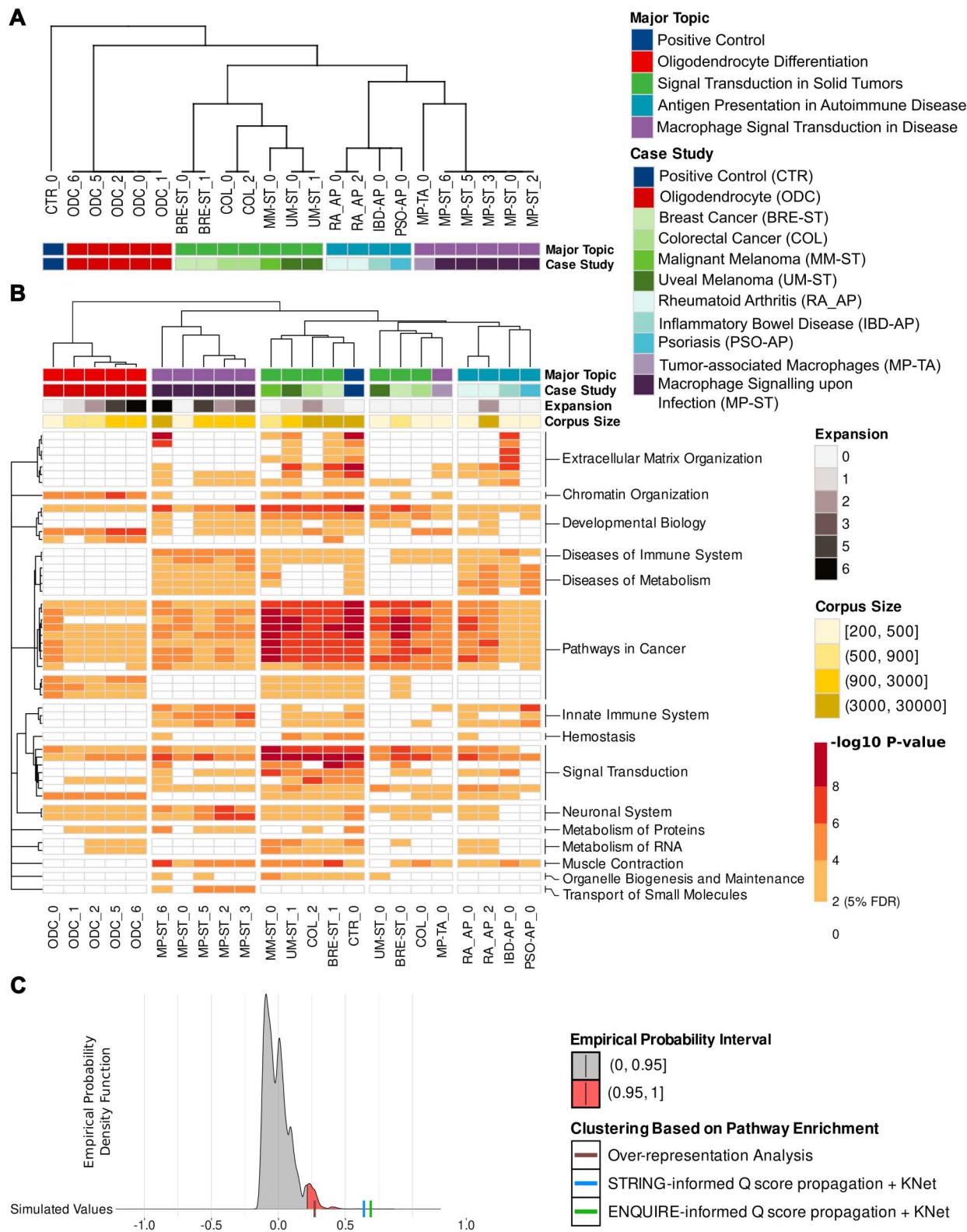

**Fig 6. ENQUIRE-generated graphs enhance the context resolution of pathway enrichment analyses. A**: reference dendrogram showcasing the expected categorization of the case studies described in Table 6. The number following a case study abbreviated name indicates the expansion counter. Network expansions that did not yield any new gene were excluded. **B**: Topology-based pathway enrichment, obtained by applying *Q*

score propagation and SANTA's KNet function on ENQUIRE-informed gene-gene associations (see Post Hoc Analyses under <u>Materials and methods</u>). The heatmap shows the unadjusted p-values for the 50 enriched Reactome pathways with at least one significant, adjusted p-value (5% FDR) and highest variance across case studies (the dendrogram was computed on the complete statistic). Pathways are clustered according to Reactome's internal hierarchy. We respectively apportioned the dendrograms into 5 and 15 partitions to visualize their respective coherence to Major Topic and Reactome Categories. Legends for expansions, rounded corpus size, and p-values ranges are provided. C: Permutation tests of Baker's gamma correlation between the reference dendrogram (**A**) and clustering obtained from alternative pathway enrichment analyses, as in **B**. Colored areas indicated probability intervals obtained from simulating correlations between reference and sampled dendrograms. See <u>Materials and methods</u> for further details.

equivalent in all tested pathways; iii) R-HSA-5663202 subpathway categorization is associated with lower p-values both globally and within the same major topic (<u>S6 Fig</u>). Perhaps unsurprisingly, we concluded that proteins from these pathways like MAP-kinases and PKB are generally involved in the explored case studies; this also suggests that the observed clustering of cancer-related studies is not exclusively dependent on the enrichment of cancer pathways. Finally, we quantitatively assess the context resolution of the ENQUIRE-informed enrichment (<u>Fig 6C</u>). To this end, we performed a permutation test on the observed Baker's gamma correlation value between dendrograms (<u>Fig 6A</u> and <u>6B</u>), which allows to statistically assess their similarity [<u>57</u>]. We benchmarked its significance against two other methods, namely gene set over-representation analysis (ORA), and topology-based pathway enrichment analysis using STRING's high-confidence functional associations, instead of ENQUIRE-generated co-occurrences, to compute the $Q$ node scores (see <u>Materials and methods</u>). All methods generated a dendrogram significantly closer than expected to the reference. In our analysis, topology-based enrichments outperform ORA, with the ENQUIRE-informed score moderately improving the performance over the STRING-informed equivalent (0.69 and 0.64, respectively). Taken together, these results suggest that ENQUIRE-generated networks can effectively represent contextual, biological differences and similarities between case study corpora. While ENQUIRE-annotated genes are sufficient for context resolution, the use of topology-based methods that incorporate corpus-specific co-occurrence information improves the performance.

## Discussion

ENQUIRE is a novel computational framework that combines text mining, network reconstruction, and literature querying, offering an alternative to manual literature curation and database mining. ENQUIRE interrelates gene and MeSH mentions through co-occurrence networks and tabulated references via a probabilistic framework that accounts for the bias of uneven representation of entities in the input literature corpus, without being dependent on the corpus size.

From the text mining perspective, we opted for a compromise between coverage of unannotated article abstracts (gene normalization) and high-fidelity, pre-computed concept annotations (MeSH retrieval). ENQUIRE's gene normalization strategy is appropriate for reconstructing co-occurrence gene networks with affordable computational requirements, and scales well with large input corpora, without the need of restricting the analysis to databases of pre-annotated gene mentions [<u>58</u>]. The combination of a curated lookup table with abstract-specific blocklists enhances precision, thus leading to co-occurrence networks with fewer false positives, compared to recall-oriented approaches like BERN2 [<u>47</u>,<u>59</u>].

A benefit of ENQUIRE is that the obtained gene/MeSH co-occurrence network can prime further information retrieval beyond text mining. Differently from previous works on gene/MeSH relations, our statistical framework is independent of the user scope (genes or MeSH

can be mined separately) and is not immutable with respect to a species or general topic (e.g. diseases) [18,60–62]. Instead, ENQUIRE automatically constructs PubMed queries from network-derived genes and MeSH to expand the input corpus, and in turn the network.

In the case study, we have shown how ENQUIRE's network-based approach enables *post hoc* analyses in conjunction with database knowledge to infer contextual gene sets, enriched molecular pathways, and to predict novel or more comprehensive gene hubs. In scenarios where a collection of genes of interest is known, such as an omics experiment, a computational model, or a case study in a molecular tumor board, by constructing queries restricted to such genes and using the resulting corpus as input ENQUIRE can enhance their biological interpretation, suggest relevant processes and components, and motivate the selection of molecular targets.

We also assessed ENQUIRE's performance using compelling case scenarios. For example, we investigated the relationship between ENQUIRE-suggested co-occurrences and database-annotated gene interactions. Our results indicate that ENQUIRE-generated gene co-occurrence networks reflect experimental and database-annotated functional gene associations. At the same time, ENQUIRE can also generate networks with previously unannotated wirings that can encourage novel exploratory analyses (Fig 5B). We also analyzed the feasibility of corroborating ENQUIRE-suggested relations by mapping co-occurrence information onto a mechanistic reference network. Since there is no generalizable method to project a network of indirect relations (co-occurrences) onto a mechanistic network [63–67], we designed a function to score a physical interaction network using ENQUIRE-generated networks. This allowed us to verify that the enriched pathways in original and expanded ENQUIRE networks reflect their contexts and enable the comparison of multiple case studies. This strategy still poses some limitations in terms of choosing a reference network and pathways to be tested. We developed a *post hoc* analysis framework as we think that text-mined co-occurrence networks can be especially valuable to inject context-specific information into general-purpose, pre-annotated relations. In these terms, the approach proposed by ENQUIRE could be beneficial for pathway curation and novel pathway discovery. However, further research is needed to investigate a robust methodology that allows the rescaling of node and edge weights of a reference, high-confidence interaction network according to context-specific co-occurrences.

We designed ENQUIRE as a series of modular, open-source components that can be combined and expanded to tune its performance. For instance, one could insert a part-of-speech recognition parser upstream of the co-occurrence detection step to strengthen its criteria [68]. Similarly, one can implement a propensity matrix into the random graph model to further weight a co-occurrence with its textual context [14,31]. Since the efficiency of deep learning models is increasing, we do not exclude the adoption of tools such as AIONER to perform biomedical entity recognition in ENQUIRE without limiting its scalability [11]. Given that gene normalization relies on the utilized lookup table of reference gene symbols and aliases, ENQUIRE's accuracy depends on how comprehensive and free of ambiguities this table is. The current version of our algorithm only performs normalization of *H. sapiens* genes and corresponding mouse orthologs. Still, it can be adapted to perform gene normalization of any other species by supplying an appropriate lookup table, such as those provided by the STRING database [69].

Our goal was to construct a robust framework for text mining, network reconstruction, automatic querying, and downstream analysis that could tackle arbitrarily complex case studies with modest resource requirements. Since the standalone version of the algorithm requires some background in computer programming, we are working on a web version of ENQUIRE to ease its adoption among biomedical researchers.

## Materials and methods

### Description of the ENQUIRE algorithm

**Extraction of article metadata.** ENQUIRE uses the NCBI's e-utilities to query and fetch information from the PubMed database [70]. *Epost* is used to request a collection of PMIDs, *efetch* to extract their metadata in XML format, and *esearch* to construct PubMed queries.

**MeSH term extraction.** For each MEDLINE-indexed, input PMID, if the MeSH entity scope is selected, ENQUIRE retrieves MeSH main headings ("descriptors") and subheadings ("qualifiers") from their respective *efetch*-retrieved XML files. These MeSH terms are further selected to match biomedically relevant, non-redundant categories, by exploiting the tree-like, hierarchical structure of the MeSH vocabulary. By default, ENQUIRE only retains members downstream of the MeSH categories A (Anatomy), C (Diseases), D (Chemicals and Drugs), and G (Phenomena and Processes), except for sub-categories G01 (Physical Phenomena), G02 (Chemical Phenomena) and G17 (Mathematical Concepts).

**Gene normalization from article abstracts.** For each input PMID, if the gene entity scope is selected, ENQUIRE retrieves article abstracts from their respective *efetch*-retrieved XML files. Additionally, the "Other Term" fields containing non-MEDLINE-indexed keywords are also *efetch*-retrieved and concatenated with the corresponding abstracts. Since other authors have shown that the proportion of gene mentions does not significantly differ between abstracts and full-body texts [71], we only mine the abstracts for gene mentions. In contrast to standard named entity recognition of genes (NER), whose task is to exactly match the character span of a gene mention, ENQUIRE's text mining framework aims at detecting at least one gene alias per unique reference gene mentioned in an abstract. We therefore designed a "Swiss cheese model" for gene normalization, in which multiple methods complement each other to improve the global precision. In brief, ENQUIRE applies up to two algorithms to each unprocessed abstract: i) the Schwartz-Hearst algorithm to detect single-word abbreviations and their respective definitions [45]; ii) the optional scispaCy model (*en_ner_jnlpba_md*) to identify words classified as "CELL_LINE" or "CELL_TYPE" [46]. This allows ENQUIRE to construct abstract-specific lists of blocked terms that discard i) ambiguous abbreviations whose definitions are not similar to any gene alias from a pre-annotated lookup table, and ii) ambiguous or unwanted mentions to cell entities containing gene aliases, such as "CD8+ T cell". Finally, a tokenization module generates potential gene-alias-matching tokens and redirects them to a unique, reference gene symbol using the lookup table.

**Construction of the lookup table of reference gene names and respective aliases.** Similar to previous approaches [72], ENQUIRE performs NER of *Homo sapiens* and *Mus musculus* gene mentions, while also redirecting the latter to their respective human homologues using MGI's mouse/human orthology table [73]. Each reference gene name corresponds to a HGNC-approved symbol [74]. Additional mouse and human gene aliases were pooled from HGNC ("previous symbols", "previous names", "alias symbols", "alias names"), ENSEMBL ("gene stable ID", "gene description", "gene name"), Uniprot ("gene names", "protein names"), and miRBase ("ID", "alias", "name") [75–77]. We manually inspected sources of ambiguities and lack of spelling variants: for example, we added miRNA names without species suffixes (e.g. "miR-335" from "hsa-miR-335"), multiple spellings for lnc- and mi-RNAs (e.g. "LNC/Lnc/lnc", "miR/mir") and removed aliases identical to common acronyms for experimental techniques (e.g. "MRI", "NMR", "TEM"). We transcribed Greek letters to their names in the Latin script. We resolved ambiguities due to aliases reported under more than one reference symbol, by either assigning the alias to a single reference, or by excluding the alias.

**Abstract tokenization for named-entity recognition of genes.** ENQUIRE mostly performs named-entity recognition of genes (NER) from article abstracts by exact matches

between gene aliases and space- or punctuation-separated word tokens. We exclude general-purpose English words annotated in the *English-words* Python library to reduce the computational burden of mapping gene mentions. Like for the lookup table, we transcribe Greek letters to their names in the Latin script. Special attention is put to hyphen- and slash-containing tokens, tracing their usage as integral parts of gene aliases (e.g. "TNF-alpha") or separators (e.g. "FcγR-TLR Cross-Talk" – PMID 31024565, "Akt/PI3K/mTOR signaling pathway" – PMID 35802302). When cases of the latter kind occur, the algorithm requires all hyphen- or slash-separated words to be gene aliases, in order to be considered individual tokens. Then, ENQUIRE tokenizes the abstract into single-word tokens and interprets unambiguous tokens as the corresponding reference gene symbol if they match an alias in the lookup table. Multiple mentions of the same gene within an abstract count as one.

**Abstract-specific lists of blocked terms using cell entity mentions and abbreviation-definition pairs.** Any token that exactly matches an alias from the lookup table is redirected to the respective reference symbol, except when either the scispaCy *en_ner_jnlpba_md* or Schwartz-Hearst models classify that same token as part of "CELL_LINE" or "CELL_TYPE" entities, or as an abbreviation. In the former exception, the token is added to a list of blocked terms and any of its mentions within the abstract text are excluded from further gene normalization steps. In the latter exception, we evaluate the validity of an alias-matching abbreviation by means of its definition, as inferred by Schwartz-Hearst. We perform string comparison to calculate alignment scores between the definition and any recorded alias of the same reference symbol matched by the abbreviation. To this end, we implemented the Needleman-Wunsch algorithm for global alignment, with match score equal to 1, gap opening and mismatch penalties equal to −1, and gap extension penalty equal to −0.5 [78]. Next, we calibrated a threshold for either retaining or discarding an alias-matching abbreviation according to its optimal alignment score. We used a dataset of abbreviation-description pairs from more than 300 abstracts and generated a distribution of scores by aligning any description to any annotated alias. Intuitively, there could only be a handful of alignments between an actual gene description and the aliases referring to that same gene, as opposed to several alignments between that same description and unrelated aliases. Therefore, we treated the above-derived distribution as a model describing false positive alignments between descriptions and gene aliases. Finally, we identified a range between 0.1 and 0.2 that respectively correspond to 95th and 99th percentiles of the distribution of alignment scores as a sensible interval for choosing the threshold. We opted for a threshold of 0.15. Therefore, for any description whose abbreviation matches a gene alias, ENQUIRE records a gene mention only if the maximal alignment score against any alias of that same gene is higher or equal to this threshold; else, the abbreviation is added to the blocklist and all of its mentions within the text are excluded. Notice that the blocklist is independently computed for each abstract, thus making ENQUIRE's gene normalization moderately adaptive with respect to syntactical context.

**Annotation of co-occurrences.** ENQUIRE records the occurrences of MeSH and gene entities within each input article. Then, it converts the recorded co-occurrences into an undirected multi-graph, where gene or MeSH terms become nodes, and each recorded co-occurrence between two entities becomes an edge. Thus, the network has as many nodes as the number of unique MeSH and gene symbols, with as many edges between two nodes as the number of PMIDs in which they co-occur.

**Reconstruction of a weighted network of significant co-occurrences.** ENQUIRE implements the soft-configuration model of Casiraghi and Nanumyan applied to undirected, unweighted edge counts to select significant co-occurrences among entities, adjusted to 1% FDR [31]. The test statistics follows a multivariate hypergeometric distribution, under the

null hypothesis of observing a random graph whose expected degree sequence correspond to the observed one. This allows us to condition the testing to the sheer, per-entity occurrence, which serves as a proxy for leveraging this kind of literature bias in the corpus. It is important to note that the null model does not assume independence of individual edges, but merely their equiprobability. This selection results in an undirected, single node-to-node edge co-occurrence graph (i.e., a simple graph).

After generating the simple graph, ENQUIRE counts significant pairwise co-occurrences by enumerating the subset of PMIDs associated to both entities in each pair. For each pair of entities $g_i$ and $g_j$ that co-occur in at least one article, we define the weights $w$ and distances $\tilde{w}$ accounting for the sheer co-occurrence $X(g_i,g_j)$ as follows:

$$w_{g_i,g_j} := \psi\Big(X\big(g_i,g_j\big),\bar{X}\Big), \quad w_{g_i,g_j} \in \big(0,1\big]$$

$$\tilde{w}_{g_i,g_j} = 1 - w_{g_i,g_j}$$

$$X\big(g_i,g_j\big) = \Big|\{P \in \mathrm{PMIDS} \,|\, g_i,g_j \in E^P\}\Big|$$

Where $X$ is the mean co-occurrence between any two entities in the corpus, $\Psi(\cdot,\bar{X})$ is the zero-truncated, Poisson cumulative density function with a lambda of $\bar{X}$, and $E^P$ is the set of all entities annotated within the PMID $P$ that belongs to the submitted *PMIDS* corpus. This scoring system assigns higher relevance to co-occurrences that appear more often than average. Notice that $\tilde{w}$ shall not be interpreted as a strict mathematical distance, but rather as a measure of closeness between connected nodes. For each pair of adjacent entities $g_i$ and $g_j$ in the simple network, we assign the weights $w_{gi,gj}$ and distances $\tilde{w}_{gi,gj}$ to their mutual edge. Additionally, we prune poorly connected nodes by modularity-based, $w$-weighted Leiden clustering [79] and removal of communities that consist of a single node. From the resulting gene/MeSH heterogeneous network, we also extract the respective gene- and MeSH-only subnetworks.

ENQUIRE-generated gene/MeSH networks can consist of multiple connected components, i.e., subgraphs. To exclude unimportant components, a subgraph $S$ is retained for subsequent computations only if the fraction of corpus articles covered by $S$ is higher than a threshold value, as formally defined in

$$T_S := \frac{\Big|\{P \in \mathrm{PMIDS} \,|\, E^P \cap E^S \neq \varnothing\}\Big|}{\big|\mathrm{PMIDS}\big|} \geq t, \quad T_S \in \big(0,1\big]$$

where $P$ denotes a PMID belonging to *PMIDS*, and $E^P$ and $E^S$ refer to the sets of gene or MeSH entities recorded in either $P$ or $S$. Therefore, $T_S$ reflects the representativeness of $S$ with respect to the entirety of the submitted corpus. The value of $t$ can be set by the user. To avoid introducing irrelevant entities, ENQUIRE stops without further network expansion if the gene/MeSH network and the respective gene- and MeSH-only subnetworks individually contain only a single, connected component with $T_S \geq t$.

Finally, we compute the weight of a node $g$ in the connected graph $S$ utilizing the composite function $W$, which is the product of normalized metrics for betweenness centrality ($b$) and $w$-weighted degree strength ($d$):

$$W\big(g,S\big) := F_b\Big(b\big(g,S\big)\Big) \cdot F_d\Big(d\big(g,S\big)\Big), \quad W \in \big(0,1\big]$$

Here, $F_x$ denotes the empirical cumulative density function for the corresponding $x$ parameter, calculated over $S$.

**Construction of communities from "information-dense" cliques.** To identify the most relevant parts of the gene/MeSH network, ENQUIRE first identifies the maximal cliques of order three or more. By definition, these are graphlets whose nodes are all adjacent to each other and not a subset of a larger clique. Applying the KNet function from the SANTA R package [52] to the gene/MeSH network having distances $\tilde{w}_{gi,gj}$, we select cliques that form significant clusters of associated entities. The permutation test procedure internal to KNet allows us to consider the network topology and adjust each maximal clique's significance, in case many other cliques of similar size exist in the network. We set the significance level for this test to 1% FDR. Subsequently, ENQUIRE generates a pruned network $C$ containing only statistically significant cliques. Here, ENQUIRE stops if the gene/MeSH network contains less than two significant cliques according to KNet. Next, ENQUIRE identifies communities in the $C$ network using modularity-based, $w$-weighted Leiden clustering. ENQUIRE stops if it detects a single community that encompasses all nodes in $C$.

**Identification of community-connecting entities.** For any two distinct communities $C_i$ and $C_j$, and for a given $k$, a parameter set by the user, we can define the set of community-connecting, weighted graphlets $\Gamma_{Ci,Cj}(V_k, L_{k-1})$ whose elements $\Gamma^n_{Ci,Cj}(V^n_k, L^n_{k-1})$ must satisfy the following properties: i) all nodes $g_i$ in the $k$-sized set $V^n_k$ belong to either $C_i$ or $C_j$; ii) both intersections between $V^n_k$ and $C_i$ or $C_j$ are non-empty; iii) the $w$-weighted, $k$-1 edges $L^n_{k-1}$ are sufficient to obtain a single connected component; iv) there is only one edge $l_{gi,gj}$ that connects nodes belonging to distinct communities.

For a given graphlet $\Gamma^n_{Ci,Cj}(V^n_k, L^n_{k-1})$, we can define the ranking function $R$:

$$R\left(\Gamma^n_{C_i,C_j}\left(V^n_k, L^n_{k-1}\right)\right) := -\log\left(\prod_{g_i \in V^n_k} W\left(g_i, \cdot\right) \prod_{l_{g_i,g_j} \in L^n_{k-1}} w_{g_i,g_j}\right), \quad R \in \mathbb{R}_{\geq 0}$$

$$V^n_k \in C_i \cup C_j, V^n_k \cap C_i \neq \varnothing, V^n_k \cap C_j \neq \varnothing$$

$$\left|\{l_{g_i,g_j} \in L^n_{k-1} \mid g_i \in C_i, g_j \in C_j\}\right| = 1$$

This allows us to rank each graphlet inside the set of community-connecting, weighted graphlets $\Gamma_{Ci,Cj}(V_k, L_{k-1})$ and, consequently, also rank the set of community-connecting entities $V^n_k$ in any graphlet $\Gamma^n_{Ci,Cj}$. The smaller the $R$, the closer two communities connected by $V^n_k$ are.

**Retrieval of new PMIDs via PubMed queries based on optimal connections.** To evaluate which genes and MeSH terms are particularly suited for querying, ENQUIRE constructs a multigraph $M$ where network communities become nodes and graphlet connections between two communities become edges, such that the value of their weights corresponds to the value of the $R$ function applied to the corresponding graphlet $\Gamma^n_{Ci,Cj}$. Edges whose weights $\rho_{Ci,Cj}$ do not fulfil the triangle inequality

$$\rho^\alpha_{C_i,C_j} \leq \rho^\beta_{C_i,C_z} + \rho^\gamma_{C_z,C_j}, \forall \, \alpha, \beta, \gamma, i, j, z \in \mathbb{N}$$

are excluded. Then, we solve the travelling salesman problem (TSP) utilizing Christofides' approximate solution as implemented in the Python package Networkx [80]. Through the visited edges, this yields an optimal path across communities and a corresponding collection of $V^n_k$ entity sets. Each selected $k$-sized set $V^n_k$ results in a PubMed query formulated via the NCBI's *esearch* utility [70]. We condition the search terms representing gene aliases and MeSH with "[Title/Abstract]" and "[MeSH Terms]", respectively, and exclude review articles from the results. By design, the "[Title/Abstract]" field also includes the "Other Term" field when used in a PubMed query. The constructed PubMed queries require a match for all the $k$ entities in

the optimal path – e.g., *"melanoma/immunology"[MeSH Terms] AND ("IL1B"[Title/Abstract] OR "interleukin 1-beta"[Title/Abstract] […]) AND […]*. If all queries involving a subset of the network communities lead to empty results, we prune all previously used edges from *M*, compute a new TSP solution, and submit newly generated queries, provided at least one entity per query belongs to such community subset. This process is repeated *A* times, where *A* is a parameter specified by the user. If at least one new PMID matches any of the constructed queries, ENQUIRE starts a new analysis from the union of new and old PMIDs; otherwise, it stops. The rationale behind merging old and new PMIDs is to account for the original corpus when computing the statistics on new co-occurrences.

## Post hoc analyses

**Context-aware gene sets.** To reconstruct contextual gene sets using gene/MeSH co-occurrence networks, we adapt network-based relational data to the method described by Khan and collaborators [81]. To this end, we first construct the inverse log-weighted similarity matrix between the gene/MeSH network nodes [82]. This metric prioritizes nodes sharing many lower degree neighbors rather than few higher degree ones. To transform this similarity matrix measure into a distance matrix, we multiply it by negative one and exponentiate it element-wise, hence obtaining smaller values the higher the similarity. After further applying a Z-score standardization, we use the R package *DynamicTreeCut* and Ward's clustering to identify initial clusters and create an initial membership degree matrix [83,84]. Finally, we detect fuzzy clusters of genes and MeSH terms by applying Fuzzy C-means clustering to the Euclidean distance matrix, using the R package *ppclust* [85]. To speed up the computation, we also offer an alternative computation of fuzzy clusters using only the first *N* PCA-derived components explaining *v* or more of the overall variance, where *v* is a parameter chosen by the user. The resulting membership degree matrix allows annotating genes with desired cluster membership degrees and extracting the linked MeSH terms to characterize the gene set. The script implementing context-aware gene set reconstruction also offers ORA on the resulting gene clusters using Reactome pathways, restricting the *universe* to the complete set of ENQUIRE-derived genes.

**Context-aware pathway enrichment analysis.** We designed a method to map any text-mined co-occurrence network *G* onto a mechanistic reference network *N* and infer context-specific enrichment of molecular pathways. With this strategy, we attempt to mechanistically explain the indirect relationships that constitute the co-occurrence network. To this end, we define the fitness score *Q* for every gene *g* in *N* with non-zero node degree *d(g,N)*:

$$Q(g) := d(g,N)^{-1} \cdot \sum_{g_i \in V(G)} \sum_{g_j \in V(G)} e^{-\tilde{\delta}_G(g_i,g_j)} \cdot 1_{\{\delta_N(g_i,g)+\delta_N(g,g_j)\leq 2, g_i \neq g_j\}}(g), \quad Q \in \mathbb{R}_{\geq 0}$$

Here, $\tilde{\delta}_G(g_i,g_j)$ and $\delta_N(g_i,g_j)$ are the $\tilde{w}$-weighted and unweighted distances from $g_i$ to $g_j$ in the graphs *G* and *N*, respectively. The indicator function □ implies that non-text-mined genes without at least two text-mined nodes as neighbors have *Q* equal to zero. We normalize all scores to decorrelate *Q* from the node degree *d(g,N)*, similarly to other approaches in network propagation [86,87]. See S5 Fig for an example of *Q* score weighting. As a mechanistic reference network, we chose STRING's (release 11.5) *H. sapiens* network of protein-coding, physically interacting genes [50]. We exclusively combined the "experimental" and "database" channels to calculate STRING's confidence score, and then pruned all edges with score below the 90th percentile. After removing zero-degree nodes, we obtain a reference, unweighted network of 9,482 nodes and 88,333 edges. Then, we calculate *Q* scores for protein-coding genes in the STRING reference network (*N*), using the ENQUIRE-generated gene network (*G*).

We test for associations between predefined gene sets and high-scoring node clusters using SANTA's KNet function [52]. KNet takes as input the STRING reference network, its nodes' $Q$ scores, and a gene set; it then tests if the latter is enriched, based on scores and graph distances of protein-coding genes belonging to both the network and the gene set. This way, we aim at capturing known experimentally or database-derived molecular interactions relevant to ENQUIRE's input literature corpus, using topology-based enrichment analysis. We test for enrichment on gene sets derived from Reactome pathways, obtained via the Reactome Graph database [49]. The offered software implementation can also return the Q-score-weighted STRING reference network in GraphML format.

## Benchmarks

**Assessment of ENQUIRE's gene normalization accuracy and performance.** We evaluated ENQUIRE's gene normalization precision and recall using abstracts from the NLM-Gene corpus mentioning at least one *M. musculus* or *H. sapiens* gene – 479 out of 550 entries [44]. We tested the four module combinations obtained by either including or excluding the cell entity recognition module *en_ner_jnlpba_md* and the Schwartz-Hearst abbreviation-definition algorithm [45,46]. We compared the computational performance of ENQUIRE's gene normalization method using both *en_ner_jnlpba_md* and Schwartz-Hearst against GNorm2 implementation of Bioformer [47,48]. We computed wall time by accounting for both text processing and loading of required data such as gene alias lookup tables and machine learning models. RAM usage was measured using resident set size (RSS) measurements returned by the Linux built-in function *ps*. We ran the computations on a Linux computer with 20 CPUs (3.1 GHz) and 252 GB of RAM. Up to 8 cores were used for parallelization.

**Inference of reactome gene sets from reference literature.** We extracted annotated genes and reference literature for all *H. sapiens* Reactome pathways from the Reactome Graph database [49]. We employed NCBI's *esearch* and *elink* utilities to retrieve primary research articles cited by review articles [70]. After excluding pathways with less than three primary literature references or only one annotated human gene, we obtained a set of 967 pathways. For each pathway literature corpus, ENQUIRE performed one network reconstruction, set to only extract gene mentions from article abstracts. We evaluated the effects of corpus size, pathway size, and average gene-gene co-occurrence per abstract on precision and recall of ENQUIRE's gene normalization and network reconstruction. We also evaluated the correlation between true positives and the corpus- and network-based node weight *W*.

**Estimate of molecular interrelations.** We automatically generated a list of case studies by crossing leaf nodes downstream of *Diseases* and *Genetic Phenomena* (G05) MeSH categories. We then constructed a PubMed query from each pair by "AND" concatenation. Examples of such queries are *"Stomach Neoplasm"[MeSH Terms] AND "Chromosomes, human, pair 18"[MeSH Terms]*, and *"Acquired immunodeficiency syndrome"[MeSH Terms] AND "Polymorphism, single nucleotide"[MeSH Terms]*. For each query result with a size between 50 and 500 articles, we executed one network reconstruction. If obtaining a gene-gene co-occurrence network, we investigated whether its set of genes produced a network with more functional interactions than expected by chance. To obtain background distributions of edge counts for each gene set size observed with ENQUIRE, we sampled one million random gene sets and cumulated their interconnecting edges in STRING's v. 11.5 *H. sapiens* functional protein network [50]. We only included functional associations from experiments, co-expression, and third-party databases with a cumulative score higher than 0.7 between proteins. The significance of each ENQUIRE-generated gene set's edge count was computed from the right-tailed probability of the empirical distribution.

Moreover, we compared the ENQUIRE-generated gene-gene wirings to STRING-derived associations using the DeltaCon similarity measure in a permutation test [51]. To this end, we generated 10,000 random graphs for each observed ENQUIRE network. Each random graph was obtained through 300 random edge-swapping attempts while preserving the degree sequence of the original network. To obtain sensible probability densities, we focused on ENQUIRE-generated networks with degree sequences allowing at least ten different realizations of a graph. We followed the formula $\Pi^n_i d_i!$, where $d_i$ is the degree of the $i$-th node of a graph containing $n$ nodes.

**Assessment of context resolution by topology-based enrichment of molecular pathways.** To show that ENQUIRE preserves context-specific molecular signatures, we designed a broad panel of case studies (Table 6). Each corpus consisted of the union of references contained in three independent reviews accessible via NCBI's *elink* utility [70]. We selected reviews from the results of PubMed search queries consisting of two or three MeSH terms (e.g., *"Melanoma"[MeSH Terms] AND "Signal Transduction"[MeSH Terms]*), favoring PubMed-ranked best matches when possible. We also included an unspecific positive control group consisting of the union of all context-specific corpora. This experimental design allowed us to construct a reference dendrogram that clusters the case studies only based on baseline biological knowledge, expecting expanded networks of a case study to cluster together with the originally reconstructed one. Then, we applied ENQUIRE with default parameters to each case study and analyzed all resulting gene-gene networks, i.e., from original and expanded corpora. We computed pairwise similarities between node and edge sets of the constructed networks using Szymkiewicz-Simpson overlap coefficient (OC):

$$OC(X,Y) = \frac{|X \cap Y|}{\min(|X|,|Y|)}, \quad OC \in [0,1]$$

Where *X* and *Y* are either two non-empty node sets or two non-empty edge sets. An OC of 0 indicates no overlap, while an OC of 1 indicates the smaller node or edge set is a subset of the larger one. By construction, same-case-study original and expanded networks possess OCs of 1 with each other. We applied the *post hoc*, context-aware pathway enrichment analysis described above to all generated networks. We tested the enrichment of Reactome pathways with sizes ranging from 3 to 100 genes, categorized as in the database's *Top-Level Pathways* and disease ontologies [49]. We performed hierarchical clustering of the networks using Euclidean distance and Kendall's correlation based on network-specific, KNet-generated p-values. We compared the resulting dendrogram to the expected one by a permutation test of Baker's gamma correlation using one million permutations of the original dendrogram [57]. We also compared the results to two alternative statistics: i) over-representation analysis of nodes from the ENQUIRE-generated networks (the collection of all genes observed in any case study was used as the "universe"); ii) KNet statistics, using *Q* scores based on STRING's high-confidence functional association network (described above) and ENQUIRE-derived gene nodes.

## Supporting information

**S1 Code. ENQUIRE's source code.** It consists of several Python and R scripts orchestrated by the main bash program. Further information is available on https://github.com/Muszeb/ENQUIRE.
(ZIP)

**S1 Fig. ENQUIRE's flowchart.** The pipeline's schematics is described with respect to start and end points (grey ellipses), input, parameters, and generated data (blue parallelograms),

algorithms (green rectangles), filtering (red triangles), pre-computed data (pink halfpipes), and branching points (yellow diamonds). NER: named-entity recognition. PMID: PubMed identifier. MeSH: Medical Subject Heading. Detailed explanation of the parameters and algorithms is provided in the main text.
(TIF)

**S2 Fig. Memory and CPU usage of a typical ENQUIRE run.** The chart shows the performance monitoring of the example ENQUIRE run described in Results and Fig 2, in which 2 expansions for a total of three iterations were performed. We used a Linux computer with 8 CPUs (2.5 GHz) and 16 GB of RAM. 6 cores were used for parallelization. Each dot represents a submodule launched by ENQUIRE, with the elapsed time at which it terminated as x-coordinate, and the maximum registered RAM usage, in the form of Resident Set Size (RSS, in megabytes), as y-coordinate. Cumulative elapsed time at the end of each reconstruction-expansion cycle is indicated. Lines in-between processes are colored by the maximum CPU usage, which is defined as the used CPU time divided by the time the process has been running, in percentage. This estimate does not typically add up to 100%. Higher CPU usage imply higher workload for each of the utilized cores. Resource usage of parallel socket cluster (PSOCK) protocol can be underestimated, as this protocol generates parallel processes whose process identifiers (PIDs) are independent of ENQUIRE's PID and not monitored. Nevertheless, ENQUIRE restricts the memory usage of PSOCK-based parallel processes, so that their aggregated memory usage is always less than 25% of the available RAM at a given time, possibly reducing the effective number of cores used.
(TIF)

**S3 Fig. Subgraph of the STRING reference, physical network showing the three InfoMap-based communities with highest average ENQUIRE-informed Q-scores.** We informed the Q-scored STRING network from ENQUIRE's context-aware pathway enrichment analysis applied to the non-expanded gene co-occurrence network for the case study *Ferroptosis and Immune System*. After applying the InfoMap community detection algorithm with normalized Q-scores as probability weights, we selected the three communities with highest average score (colored areas) and pruned the original STRING network using the union of their corresponding protein-coding genes. Nodes with non-zero Q-score are labeled and colored in magenta.
(TIF)

**S4 Fig. Diversity in nodes and edges from reconstructed and expanded networks generated by ENQUIRE.** We computed similarity measures between ENQUIRE-inferred, co-occurrence gene networks based on the case studies described in Table 6. The number following a case study abbreviated name indicates the expansion counter. Network expansions that did not yield any new gene were excluded. Panel **A** depicts similarities between the networks' node sets, while panel **B** depicts similarities between edge sets. Numbers and color gradient report Szymkiewicz-Simpson overlap coefficient percentages (OC). An OC of 0% indicates no overlap, while an OC of 100% indicates the smaller node or edge set is a subset of the larger one. By construction, same-case-study original and expanded networks possess OCs of 100% with each other. OC between the positive control (CTR) and other case study networks are highlighted in red.
(TIF)

**S5 Fig. Example of Q score weighting.** The top row shows three simulated co-occurrence networks $G$ with the same set of text-mined genes (squares), generated with progressively higher edge-forming probability, and sampling edge weights $\tilde{w}$ from a uniform distribution

in *[0,1]*. Genes from an immutable reference network *N* containing both text-mined and non-text-mined genes (circles) are weighted by the *Q* score. For each gene *g* in *N*, its weight *Q* is a function of the text-mined genes in the *g*-neighbourhood and their $\tilde{w}$-weighted distances in the network *G*. Nodes with relatively more connections to text-mined nodes in the reference network possess higher *Q* scores, irrespective of being text-mined or having a high node degree. See the non-text-mined node Y and the text-mined node J as an example.
(TIF)

**S6 Fig. Constitutively enriched subpathways of *Diseases of signal transduction by growth factor receptors and second messengers* (R-HSA-5663202). A**: differences in network distances between genes belonging to R-HSA-5663202 subpathways and other Reactome pathways, based on STRING's reference physical network (FDR-adjusted p-value = 0.27, Mann-Whitney U test). The binned network distances are used by KNet to compute a topology-based pathway enrichment. **B**: differences in Spearman correlations between KNet p-values and network size, in R-HSA-5663202 subpathways and other Reactome pathways (FDR-adjusted p-value = 0.79, Mann-Whitney U test). **C**: differences in Spearman correlations between KNet p-values and corpus size, in R-HSA-5663202 subpathways and other Reactome pathways (FDR-adjusted p-value = 0.23, Mann-Whitney U test). **D**: differences in p-value distributions between R-HSA-5663202 subpathways and other pathways, across all case studies (FDR-adjusted p-value = $6.5 \cdot 10^{-5}$, mixed model ANOVA). **E**: differences in p-value distributions between R-HSA-5663202 subpathways and other pathways, for each major topic (FDR-adjusted p-value (Positive Control) = 0.04– Mann-Whitney U test, FDR-adjusted p-value (Oligodendrocyte Differentiation) = $1.3 \cdot 10^{-2}$, FDR-adjusted p-value (Signal Transduction in Solid Tumors) = $1.4 \cdot 10^{-4}$, FDR-adjusted p-value (Antigen Presentation in Autoimmune Diseases) = $2.3 \cdot 10^{-5}$, FDR-adjusted p-value (Macrophage's Signal Transduction in Disease) = $3.9 \cdot 10^{-4}$– mixed model ANOVA). See https://zenodo.org/records/12734778 for details on the test statistics.
(TIF)

## Acknowledgments

We thank Martin Eberhardt, Christopher Lischer, Jimmy Retzlaff, Esther Güse, and Suryadipto Sarkar for the useful scientific discussions, comments on the manuscript, and testing the installation and running of the algorithm. We also want to thank Jacopo Innocenti for lending his arts talents to the graphical abstract available on GitHub.

## Author contributions

**Conceptualization:** Luca Musella, Julio Vera.

**Data curation:** Luca Musella, Max Widmann.

**Formal analysis:** Luca Musella, Max Widmann.

**Funding acquisition:** Julio Vera.

**Investigation:** Luca Musella, Julio Vera.

**Methodology:** Luca Musella.

**Project administration:** Julio Vera.

**Resources:** Julio Vera.

**Software:** Luca Musella, Alejandro Afonso Castro, Max Widmann.

**Supervision:** Julio Vera.

**Validation:** Luca Musella, Alejandro Afonso Castro, Max Widmann.

**Visualization:** Luca Musella.

**Writing – original draft:** Luca Musella, Julio Vera.

**Writing – review & editing:** Luca Musella, Alejandro Afonso Castro, Xin Lai, Julio Vera.

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
