## [Decision Letter · Decision Letter 0]

28 May 2024

Dear Musella,

Thank you very much for submitting your manuscript "ENQUIRE automatically reconstructs, expands, and drives enrichment analysis of gene and MeSH co-occurrence networks using context-specific biomedical literature" for consideration at PLOS Computational Biology.

As with all papers reviewed by the journal, your manuscript was reviewed by members of the editorial board and by several independent reviewers. In light of the reviews (below this email), we would like to invite the resubmission of a significantly-revised version that takes into account the reviewers' comments.

The reviewers raise points related to providing more context for this importance and utility of the software, comparison to other approaches, and accounting for co-occurrence networks. The revised manuscript should address these points.

We cannot make any decision about publication until we have seen the revised manuscript and your response to the reviewers' comments. Your revised manuscript is also likely to be sent to reviewers for further evaluation.

Sincerely,

Christos A. Ouzounis

Academic Editor

PLOS Computational Biology

Stacey Finley

Section Editor

PLOS Computational Biology

Reviewer's Responses to Questions

**Comments to the Authors:**

Reviewer #1: ## Review summary

The authors create an analytical framework and a tool called ENQUIRE that utilizes an innovative combination of knowledge technologies to reconstruct and expand co-occurrence networks of biomedical concepts and gene symbols from PubMed. The presented approach can help with biological interpretation as well as generating new research hypotheses.

Quality: The quality of the presented approach is excellent. The approaches and techniques used are well-known but are presented and combined in an innovative way. Performance evaluation and result validation are presented in an exemplary manner.

Clarity: The paper is written in a clear manner. The introduction provides clear motivation. The manuscript is primarily methodological, so I suggest that the Methods section directly follows the Introduction. The list of references is relevant and adequate. I missed one or two paragraphs about previous works and state-of-the-art approaches. The provided programming code is well documented and enables the reader to replicate the results of the study.

Originality and Significance: Although the discussed problem has been widely studied in the past two decades, the presented approach is unique and presents a certain degree of novelty. Overall, the work provides an incremental application of existing methods and advances the interpretation of biological networks.

## Major issues

Dictionary-based extraction of gene names and symbols from free abstract texts can have a significant impact on the quality of downstream tasks' results. Some time ago, PubMed introduced the "Other Term" field, and it would be useful to include this as background knowledge as well.

As per the definition, an edge in a co-occurrence network signifies the strength of linkage between two knowledge concepts (i.e., MeSH terms), whereas an edge in a semantic network reveals the nature of the relationship (e.g., "MeSH Term 1 (e.g., substance) inhibits/stimulates Gene A," where the MeSH term could potentially indicate substance). I do not expect to reframe the proposed approach, but the manuscript would benefit from discussing the benefits and limitations of the presented co-occurence-based approach in relation to more novel, knowledge-based approaches with labels on edges.

Please add a paragraph or two to describe state-of-the-art approaches.

Thank you for this great paper!

Reviewer #2: the review is uploaded as an attachment.

**Have the authors made all data and (if applicable) computational code underlying the findings in their manuscript fully available?**

Reviewer #1: Yes

Reviewer #2: Yes

PLOS authors have the option to publish the peer review history of their article (what does this mean?). If published, this will include your full peer review and any attached files.

Reviewer #1: No

Reviewer #2: No
---

## [Decision Letter · Decision Letter 1]

20 Dec 2024

Dear Musella,

We are pleased to inform you that your manuscript 'ENQUIRE automatically reconstructs, expands, and drives enrichment analysis of gene and MeSH co-occurrence networks from context-specific biomedical literature' has been provisionally accepted for publication in PLOS Computational Biology.

Best regards,

Christos A. Ouzounis

Academic Editor

PLOS Computational Biology

Stacey Finley

Section Editor

PLOS Computational Biology

Reviewer's Responses to Questions

**Comments to the Authors:**

Reviewer #1: The authors significantly improved the manuscript and resolved all concerns and open questions.

**Have the authors made all data and (if applicable) computational code underlying the findings in their manuscript fully available?**

Reviewer #1: Yes

PLOS authors have the option to publish the peer review history of their article (what does this mean?). If published, this will include your full peer review and any attached files.

Reviewer #1: No

---

## [Editor Report · Acceptance letter]

PCOMPBIOL-D-24-00495R1

ENQUIRE automatically reconstructs, expands, and drives enrichment analysis of gene and MeSH co-occurrence networks from context-specific biomedical literature

Dear Dr Musella,

I am pleased to inform you that your manuscript has been formally accepted for publication in PLOS Computational Biology. Your manuscript is now with our production department and you will be notified of the publication date in due course.

With kind regards,

Anita Estes
